# Membrane-binding and activation of LKB1 by phosphatidic acid is essential for development and tumour suppression

Giada Dogliotti[1,*], Lars Kullmann[1,2,*], Pratibha Dhumale[3,4], Christian Thiele[1], Olga Panichkina[1], Gudrun Mendl[1], Roland Houben[5], Sebastian Haferkamp[6], Andreas W. Püschel[3,4] & Michael P. Krahn[1,2]

The serine/threonine kinase LKB1 regulates various cellular processes such as cell proliferation, energy homeostasis and cell polarity and is frequently downregulated in various tumours. Many downstream pathways controlled by LKB1 have been described but little is known about the upstream regulatory mechanisms. Here we show that targeting of the kinase to the membrane by a direct binding of LKB1 to phosphatidic acid is essential to fully activate its kinase activity. Consequently, LKB1 mutants that are deficient for membrane binding fail to activate the downstream target AMPK to control mTOR signalling. Furthermore, the *in vivo* function of LKB1 during development of *Drosophila* depends on its capacity to associate with membranes. Strikingly, we find LKB1 to be downregulated in malignant melanoma, which exhibit aberrant activation of Akt and overexpress phosphatidic acid generating Phospholipase D. These results provide evidence for a fundamental mechanism of LKB1 activation and its implication *in vivo* and during carcinogenesis.

[1] Molecular and Cellular Anatomy, University of Regensburg, Universitätsstr. 31, 93053 Regensburg, Germany. [2] Internal Medicine D, University Hospital of Münster, Domagkstr. 3, 48149 Münster, Germany. [3] Institute for Molecular Cell Biology, University of Münster, Schlossplatz 5, 48149 Münster, Germany. [4] Cells-in-Motion Cluster of Excellence, University of Münster, D-48149 Münster, Germany. [5] Department of Dermatology, Venereology and Allergology, University Hospital Würzburg, Josef-Schneider-Str. 2, 97080 Würzburg, Germany. [6] Institute of Dermatology, University Hospital Regensburg, Franz-Josef-Strauss-Allee 11, 93053 Regensburg, Germany. * These authors contributed equally to this work. Correspondence and requests for materials should be addressed to M.P.K. (email: Michael.Krahn@ukmuenster.de).

The serine/threonine kinase LKB1 is ubiquitously expressed and highly conserved throughout evolution. It has been demonstrated to function as a 'master kinase' potentially activating several downstream kinases[1]. Apart from its implication in carcinogenesis LKB1 plays a role in various cellular signalling pathways such as Wnt-, TGFβ-signalling or the mTOR-pathway reviewed by Vaahtomeri and Makela[2]. The latter one is controlled by LKB1-mediated activation of AMP-dependent kinase (AMPK), which is essential for cell survival and polarity in Drosophila and vertebrates, in particular under energetic stress[3–7]. Mechanistically, AMPK phosphorylates (among others) Raptor (a core component of mTOR complex 1) and TSC2 (an mTOR inhibitor), resulting in reduced mTOR activity[8–15]. Consequently, the LKB1-AMPK-axis is believed to be a key modulator in carcinogenesis and cell polarity[16].

Apart from its function in cell proliferation and tumour suppression, LKB1 is directly implicated in the establishment and maintenance of cell polarity in different cell types and organisms[17].

Although many downstream pathways mediating the function of LKB1 have been described, little is known about the upstream mechanisms regulating LKB1 activity. Two pseudokinases tightly control the localization and kinase activity of LKB1: STRADα (Ste20-like kinase (Stlk) in Drosophila) and Mo25 form a stable ternary complex with LKB1 (ref. 18). Both proteins enhance the export of LKB1 from the nucleus into the cytoplasm and increase its kinase activity[18,19]. Secondly, the conserved C-terminus of LKB1 is farnesylated in vivo and thereby might directly interact with the plasma membrane to attach the protein to the cell cortex. Although the majority of the protein accumulates at the plasma membrane of polarized (epithelial) cells, farnesylation has been reported to be not essential for the (tumour suppressor) function of LKB1 in mammalian cells or mice but might be essential for oogenesis in Drosophila[20–22]. Therefore the question remains, whether membrane association of LKB1 is essential for the kinase activity and function of the protein during development and tumour suppression.

Here we report that binding of LKB1 to membranes by direct interaction with phospholipids, in particular to phosphatidic acid, is essential for its function in vivo during development of Drosophila. Membrane association of LKB1 is required for its kinase activity and for efficient activation of AMPK in cultured mammalian cells, thus contributing to the tumour suppressor function of LKB1. Strikingly, we reveal a strong correlation between overexpression of Phospholipase D (PLD), which increases cellular levels of phosphatidic acid, downregulation of LKB1 and enhanced activity of mTOR in malignant melanoma, thus likely contributing to the pathogenesis of malignant melanoma.

## Results

### LKB1 localizes to the cortex of epithelial cells and neuroblasts.
The activity of kinases can be modulated for instance by directly influencing their enzymatic activity (e.g., by a conformational change via phosphorylation) or by targeting the protein to different subcellular compartments. In cultured mammalian cells, LKB1 accumulates in the nucleus in many cell lines and only a minor fraction of the protein is found in the cytoplasm or at the cytocortex[23–25]. LKB1 has been demonstrated to shuttle from the cytoplasm into the nucleus and back, with its two co-factors, STRADα and Mo25 enhancing cytoplasmic localization and activating the kinase[18,19,26].

Whereas endogenous LKB1 accumulates predominately in the nucleus of non-transformed fibroblasts (Fig. 1a), epithelial cells (Madin Darby Canine Kidney, MDCK) exhibit a staining of endogenous LKB1 exclusively at the cell-cell contacts, partly

overlapping with the AJ- and TJ markers E-Cadherin and ZO-1 (Fig. 1b and ref. 27). Similar, endogenous LKB1 localizes to the (lateral) membrane in epithelial cells in vivo (colon or salivary glands) (Fig. 1c,d). Remarkably, polarized epithelial cells in culture (MDCK) and in situ (colon and salivary gland) do not exhibit nuclear staining of LKB1.

In Drosophila cells endogenous LKB1 is also not found in the nucleus but localizes at the cortex of female germ line cells as well as at the lateral membrane in epithelial cells[22], whereas it shows a diffuse cytoplasmic pattern in neural stem cells of Drosophila larval neuroblasts (NBs)[28]. Similar, the Caenorhabditis elegans orthologue, PAR-4, regulating asymmetric cell division of the zygote, localizes to the entire cell cortex[29]. To test which mechanisms target LKB1 to the cortex, we raised an antibody against LKB1 and confirmed that in epithelial cells of the embryonic epidermis, endogenous LKB1 is localized laterally, co-staining with α-spectrin (Fig. 1e), but also overlapping with the zonula adherens (ZA), marked by Drosophila E-Cadherin (DE-Cad) (Fig. 1e). Notably, in embryonic NBs, we detect a clear cortical LKB1 staining in interphase as well as during mitosis (Fig. 1f–h). However, in contrast to key regulators of asymmetric cell division like the apical localized Bazooka (Baz) protein and the basally segregated adaptor protein Miranda (Mir), LKB1 does not show a polarized distribution during mitosis (Fig. 1f–h). Thus, LKB1 localizes predominately to the (lateral) plasma membrane in polarized epithelial cells and NBs in Drosophila and mammals.

### Farnesylation of LKB1 is not essential for its localization.
The C-terminus of LKB1 can be farnesylated[21,30], establishing a putative membrane targeting domain—however, this modification does not seem to be important for its tumour suppressor function in mammals[20] and for viability of mice[21]. In the Drosophila germ line, a block of farnesylation leads to a weaker cortical association and disturbed oocyte polarity[22]. To test whether this is also true for epithelial cells and NBs, we used the endogenous LKB1 promoter to express a GFP-tagged LKB1 wild-type protein or an LKB1 version with a Cys564-Ala substitution in the fly (farnesylation-deficient LKB1, lkb1::GFP-LKB1$_{C564A}$). Wild-type GFP-LKB1 is expressed at similar protein level as endogenous LKB1 (Supplementary Fig. 1a) and localizes to the lateral cortex of epithelial cells and the cortical membrane in NBs indistinguishable from the endogenous protein (Fig. 1i,k). Surprisingly, in the embryonic epidermis and embryonic NBs, GFP-LKB1$_{C564A}$ shows the same subcellular localization as its wild-type counterpart (Fig. 1j,l). The farnesylation-deficient protein shows a more cytosolic distribution only in epithelial cells surrounding the oocyte (follicular epithelium) although a substantial fraction of the protein is still associated with the lateral membrane (Fig. 1n compared to wild-type GFP-LKB1 in Fig. 1m).

Notably, LKB1$_{C564A}$ expressed from its endogenous promoter is able to rescue an lkb1-null allele (lkb1$^{X5}$) to a large extent (52% surviving flies in comparison to 69% for wild-type LKB1, Fig. 1o), indicating that farnesylation of LKB1 is not essential for the function of the protein in vivo.

### LKB1 directly binds to phospholipids.
In order to further elucidate the targeting of LKB1 to the plasma membrane, we used Schneider R+ (SR+) cells as they do not exhibit an intrinsic polarity and do not express transmembrane proteins like DE-Cad, Crumbs or Echinoid, qualifying them as a model for the analysis of direct plasma membrane targeting. As expected, GFP-LKB1 localizes to the plasma membrane in transfected S2R+ cells (Fig. 2c). In contrast, the farnesylation motif alone (GFP fused to

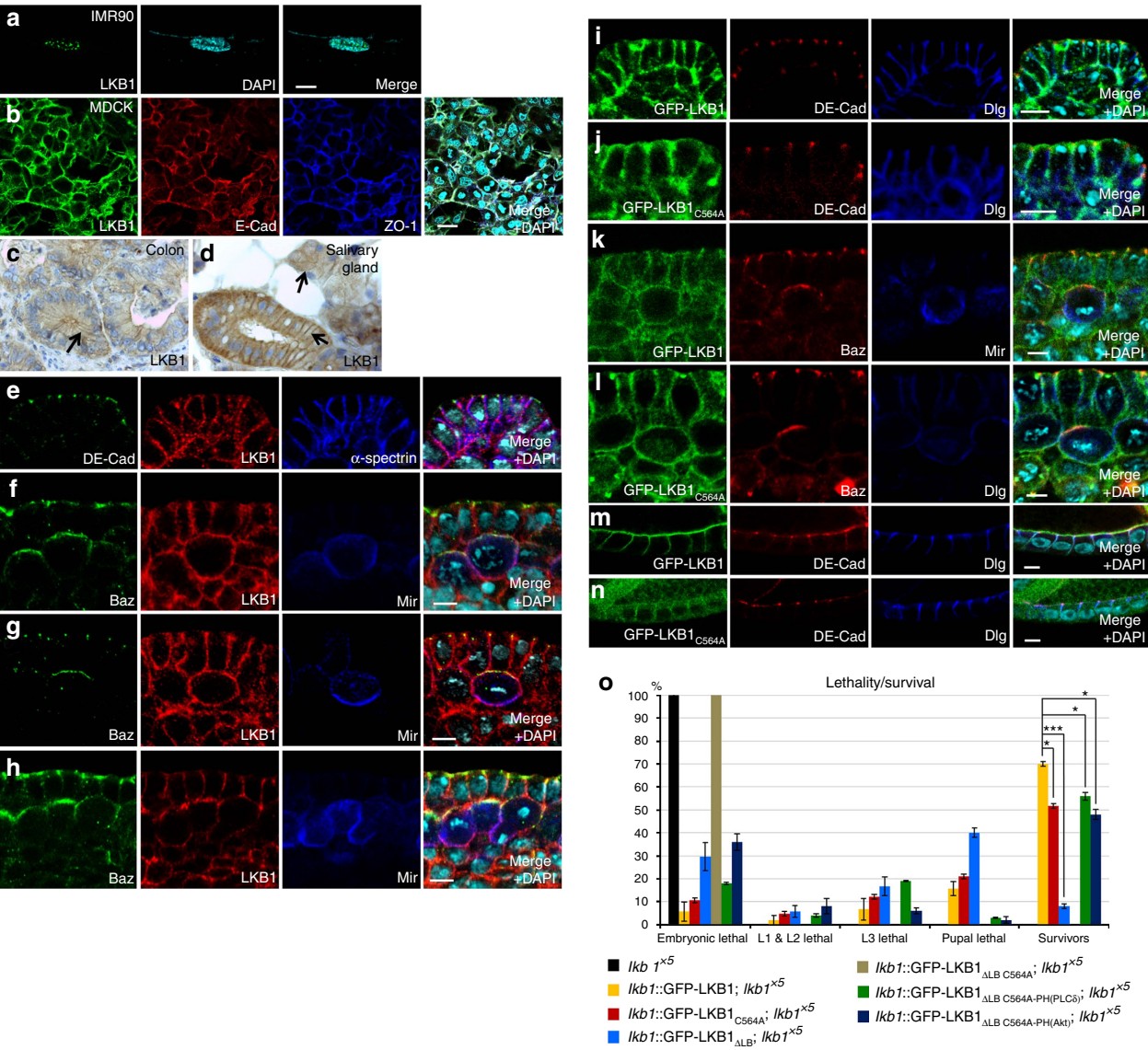

**Figure 1 | LKB1 localizes to the plasma membrane in polarized epithelial cells and neuroblasts.** (**a**) In non-transformed mammalian fibroblasts (IMR90 cells), endogenous LKB1 accumulates mostly in the nucleus. (**b**) In polarized epithelial cells (MDCK), LKB1 is targeted to the cell-cell contacts, partly colocalizing with E-Cadherin (E-Cad) and Zonula occludens protein 1 (ZO-1). (**c,d**) Sections of paraffin-embedded colon (**c**) and salivary gland tissues (**d**) show a localization of LKB1 at the lateral plasma membrane in situ (arrows). (**e–h**) LKB1 localizes to the lateral plasma membrane in epithelial cells of the *Drosophila* embryonic epidermis (**e**) and to the cortex of neuroblasts (**f–h**). (**i–n**) Wild-type GFP-LKB1 as well as farnesylation deficient LKB1 (LKB1$_{C564A}$) expressed from its endogenous promoter localize correctly to the (lateral) membrane of epithelial cells of the embryonic epidermis (**i,j**), of the follicular epithelium (**m,n**) and of neuroblasts (**k,l**). (**o**) Lethality tests of LKB1 variants as described in the methods section. Scale bars are 20 μm in **a–d**, 5 μm in **e–n**. Experiments were performed in triplicates. Error bars represent s.d. and statistical significance was determined using ANOVA: $P < 0.0001$, ***$P < 0.01$, *$P > 0.05$, not significant (NS).

the last 15 amino acids of LKB1, GFP-LKB1$_{552-C}$) is not sufficient to target the protein to the cortex (Fig. 2d), which is in line with the hypothesis that stable membrane binding requires protein modifications (e.g., palmitoylation) or membrane binding domains[31] in addition to conjugation with farnesyl acid. In contrast, a longer C-terminal fragment of LKB1 (LKB1$_{512-C}$ and LKB1$_{536-C}$) exhibits a robust cortical localization apart from a nuclear staining (Fig. 2e; Supplementary Fig. 2a). Furthermore, in embryos, a substantial fraction of GFP-LKB1$_{512-C}$ localizes to the lateral membrane (Fig. 2g).

Membrane association can be achieved not only by the interaction with transmembrane or membrane associated proteins but also by the direct binding to the lipid bilayer.

Several protein domains are known to facilitate protein–lipid interactions. Beside larger domains, polybasic motifs have been described for several proteins to bind to (phospho)-lipids, in particular to phosphatidic acid[32]. Indeed, a C-terminal polybasic motif (aa 539–551, termed lipid-binding motif = LB, Fig. 2a) facilitates a direct binding of DmLKB1 to PA, PtdIns(5)P, PtdIns(3,4,5)P3 and PtdIns(4,5)P2 as determined by a lipid overlay assay (Fig. 2b). A strong binding to PA (and to some extent to PtdIns(3,4,5)P3 and PtdIns(4,5)P2) was confirmed in liposome flotation assays (Supplementary Fig. 2b). Mutation of this motif (mutation of Arg and Lys in LKB1-LB to Ala, LKB1$_{\Delta LB}$) abolished liposome association in vitro (Supplementary Fig. 2b) as well as membrane association in cultured cells

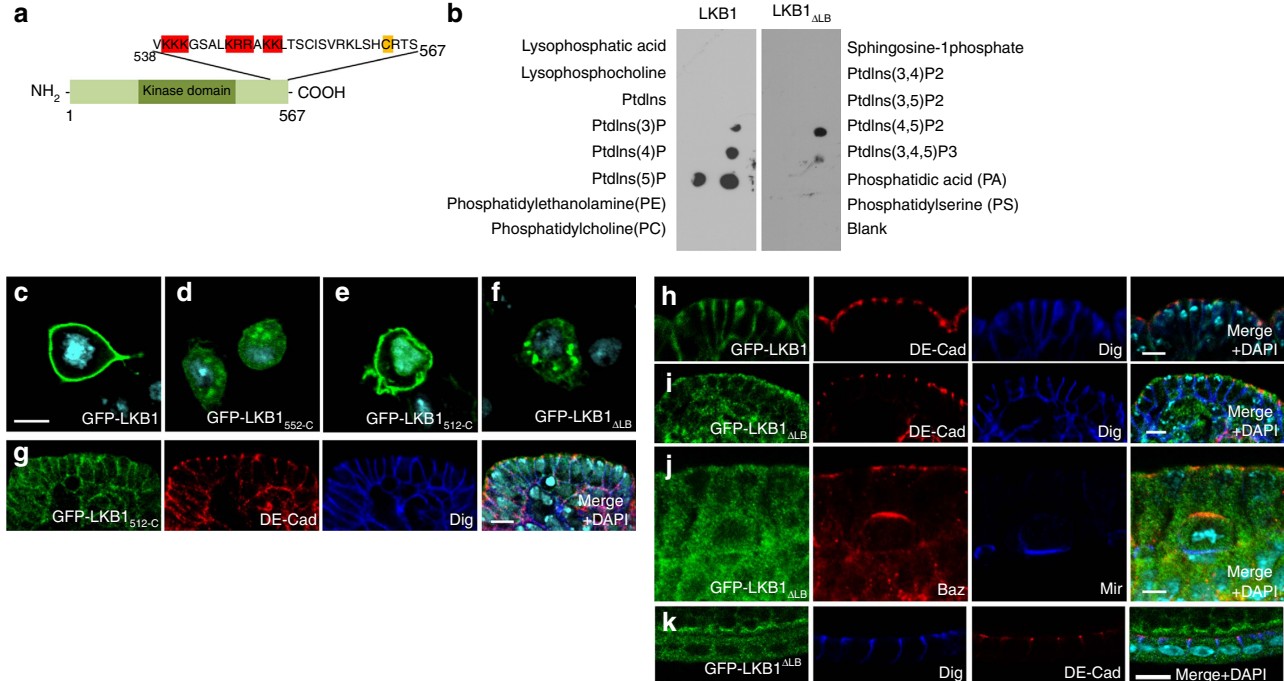

**Figure 2 | LKB1 is recruited to the lateral plasma membrane by direct binding to phospholipids.** (**a**) Schematic drawing showing *Drosophila* LKB1 with basic residues and the farnesylation motif within its C-terminal region highlighted. (**b**) Lipid overlay assay of recombinant LKB1 and LKB1$_{\Delta LB}$ ( = LKB1$_{K546A}$ $_{R547A\ R548A\ K550A\ K551A\ K539A\ K540A\ K541A}$) using diverse lipids spotted on nitrocellulose membrane. (**c–f**) GFP-LKB1 and GFP-LKB1$_{512-C}$ localize to the cell cortex of transfected S2R cells (**c,e**), whereas a fusion protein of the last 16aa of LKB1 with GFP (GFP-LKB1$_{552-C}$) or GFP-LKB1$_{\Delta LB}$ does not (**d,f**). (**g–j**) GFP-LKB1$_{512-C}$ accumulates at the lateral membrane of epithelial cells in the embryonic epidermis (**g**), similar to full length GFP-LKB1 (**h**), whereas GFP-LKB1$_{\Delta LB}$ exhibits a cytosolic localization in epithelial cells of the embryonic epidermis (**i**) and of the follicular epithelium (**k**) as well as in neuroblasts (**j**). Scale bars are 5 μm.

(Fig. 2f) and in *Drosophila* epithelia and neural stem cells (Fig. 2h–k).

**Membrane-bound LKB1 is crucial for *Drosophila* development.** LKB1 has been implicated in various cellular functions, among them cell proliferation control, suppression of tumour growth and regulation of cell polarity in various cell types[2,17]. To address the question whether membrane targeting of LKB1 is crucial for its physiological function, we performed rescue experiments with GFP-LKB1 expressed from its endogenous promoter. As indicated above, not only wild-type LKB1 but also LKB1$_{C564A}$ can rescue the embryonic lethality of maternal and zygotic mutant flies to a substantial extent, indicating that wild-type and farnesylation-deficient GFP-LKB1 can substitute endogenous LKB1 in all tissues (Fig. 1o). Notably, GFP-DmLKB1$_{\Delta LB}$ still exhibits a residual rescue capacity (8% of surviving flies in contrast to 69% for wild-type DmLKB1, Fig. 1o), which might be due to a transient or weak membrane binding mediated by the farnesylation anchor. Indeed, removal of the farnesylation motif in GFP-LKB1$_{\Delta LB}$ (GFP-LKB1$_{\Delta LB\ C564A}$) totally abolishes the rescue capacity of the mutant protein and all mutant flies die during embryonic stages like the null allele *lkb1$^{X5}$* (Fig. 1o). Moreover, *lkb1*-mutant flies expressing GFP-LKB1$_{\Delta LB\ C564A}$ exhibit the same polarity defects as the null allele alone (e.g., disruption of the anterior–posterior polarity of ovaries, Fig. 3c, compared with wild-type LKB1 rescue in **a** and *lkb1*-mutant phenotype in **b**).

Western blot analyses of lysates from embryos expressing LKB1 variants from the endogenous promoter reveal that GFP-LKB1$_{\Delta LB\ C564A}$ is unstable or rapidly degraded (Fig. 3d),

indicating that the association of LKB1 with membranes is essential for protein stability. In contrast, transient membrane association of LKB1$_{\Delta LB}$, which is probably mediated by the farnesylation anchor, is sufficient for protein stabilization, as no differences in protein levels are detectable between wild-type LKB1 and lipid-binding-deficient LKB1 (Fig. 3d).

However, overexpression of GFP-DmLKB1$_{\Delta LB\ C564A}$ in *lkb1*-mutant flies using the UAS/GAL4 system instead of the endogenous *lkb1*-promoter does not result in detectable rescue capacity (Supplementary Fig. 3a). Similar, the overexpression phenotype (rough eye formation) of LKB1 is abolished in GFP-DmLKB1$_{\Delta LB\ C564A}$ similar to a kinase dead version (Fig. 3e). Thus, we assume that it is very unlikely that impaired functionality of GFP-DmLKB1$_{\Delta LB\ C564A}$ is only due to the instability of the protein.

To further substantiate this assumption, we fused two different heterologous membrane-binding domains to the C terminus of GFP-LKB1$_{\Delta LB\ C564A}$: The PH-domain of human phospholipase Cδ (ref. 33) that preferentially binds to PtdIns(4,5)P2 (resulting in GFP-LKB1$_{\Delta LB\ C564A-PH(PLC\delta)}$), but is also capable of binding to PA with a high affinity[34], and the PH-domain of human Akt1, (GFP-LKB1$_{\Delta LB\ C564A-PH(Akt1)}$) with a high affinity to PtdIns(3,4,5)P3 (ref. 35). Similar to PLCδ, Akt1 has recently been demonstrated to bind PA *in vitro*[36]. Strikingly, both chimeric proteins stabilize the protein expression of GFP-LKB1$_{\Delta LB\ C564A}$ (Fig. 3d), localize at least partly to the lateral plasma membrane (Supplementary Fig. 4a,b) and are capable of rescuing the *lkb1* null allele to a large extent (56 and 48% respectively, Fig. 1o). These results confirm our hypothesis that membrane association of LKB1 is crucial for its function *in vivo*.

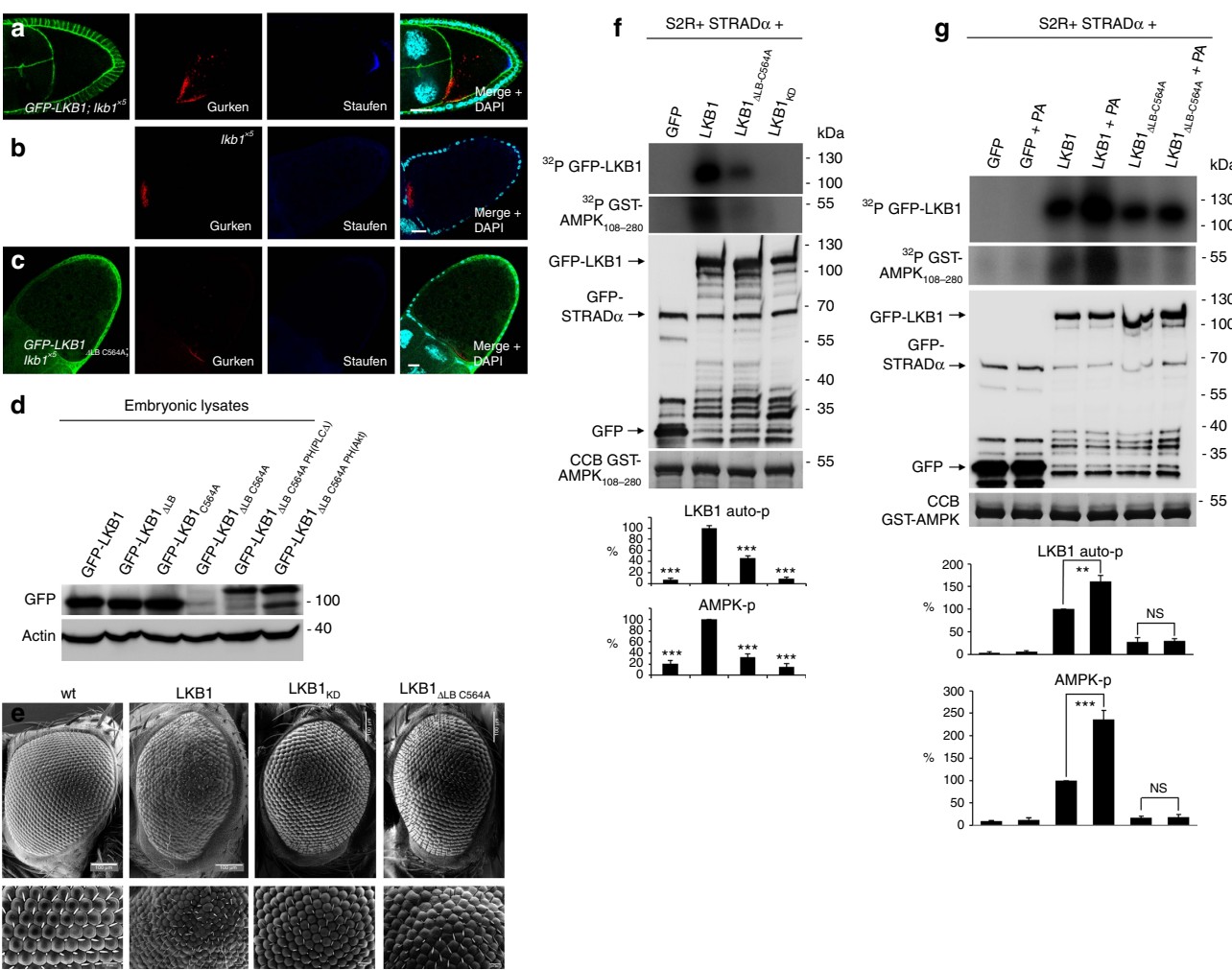

**Figure 3 | Membrane-association of LKB1 is essential for its kinase activity and function *in vivo*.** (**a–c**) Immunostainings of *lkb1*-mutant ovaries (**b**) exhibit a disturbed localization of Staufen, which is rescued by expression of GFP-LKB1 (**a**) but not of GFP-LKB1$_{\Delta LB\ C564A}$ (**c**). (**d**) Immunoblotting of LKB1 variants expressed in embryos demonstrating that the protein stability of GFP-LKB1$_{\Delta LB\ C564A}$ is strongly reduced. (**e**) Overexpression of LKB1, but not of a kinase-dead version of LKB1 (LKB1$_{D317A}$ = LKB1$_{KD}$) or GFP-LKB1$_{\Delta LB\ C564A}$ results in an impaired eye morphology ('rough eye' phenotype). (**f,g**) *In vitro* kinase assays of GFP-LKB1/GFP-STRAD purified from transfected S2R cells demonstrate a strong decrease in the kinase activity of LKB1$_{\Delta LB\ C564A}$ measured by autophosphorylation ($^{32}$P GFP-LKB1) and phosphorylation of recombinant AMPK ($^{32}$P GST-AMPK$_{108-280}$) (**f**). The addition of PA-enriched liposomes (as described in materials and methods) increased the kinase activity of wild type but not of membrane-binding deficient LKB1 (**g**). Activity of GFP-LKB1 was set as 100%. Inputs were visualized by Western Blot against GFP (GFP, GFP-Stlk and GFP-LKB1) and by Coomassie Brilliant Blue staining (CCB, GST-AMPK$_{108-280}$) Scale bars are 5 µm in **c–k** and **m**, 100 µm in **o** and **p**, 10 µm in **l** and **n**. Experiments were performed in triplicates. Error bars represent s.d. and statistical significance was determined using ANOVA: $P < 0.001$, \*\*\*$P < 0.01$, \*\*$P < 0.05$, not significant (NS).

To test whether membrane association is essential for the kinase function of LKB1, we performed *in vitro* kinase assays using the GFP-DmLKB1/STRADα complex purified from transfected S2R cells. Indeed GFP-DmLKB1$_{\Delta LB\ C564A}$ exhibits a strongly decreased kinase activity (46% in autophosphorylation and 30% in AMPK-phosphorylation relative to wild-type GFP-LKB1, Fig. 3f). Furthermore, addition of PA-enriched liposomes to GFP-DmLKB1/STRADα strongly increases the kinase activity of the LKB1 complex (autophosphorylation 1.6-fold and AMPK-phosphorylation 2.4-fold (Fig. 3g).

Reduced kinase activity of LKB1$_{\Delta LB\ C654A}$ is neither due to impaired association with the canonical binding partners STRADα or Mo25, which enhance LKB1 activity, nor due to decreased substrate binding (Supplementary Fig. 3b). Therefore, these data suggest that the lipid binding capacity of DmLKB1 is essential for its kinase function.

**The function of human LKB1 depends on membrane binding.** As association with the membrane is essential for LKB1 to accomplish its function during development of *Drosophila*, we further investigated whether this mechanism is conserved throughout evolution and controls the activity of human LKB1 (hLKB1). Similar to its fly homologue, mutation of the polybasic region at the C terminus of hLKB1 together with mutation of the farnesylation motif (Fig. 4a) resulted in an impaired membrane localization in HeLa cells (Fig. 4b,c).

Activation of AMPK is one of the most important functions of LKB1 during tumour progression, which is further enhanced under energetic stress. As activated AMPK directly and indirectly inhibits mTOR[8–15], the LKB1-AMPK pathway controls mTOR activity, e.g., upon aberrant activation of Akt, which occurs in various types of tumours with loss of function of PTEN or PI3-Kinase gain of function mutations[37]. Loss of membrane binding capacity of

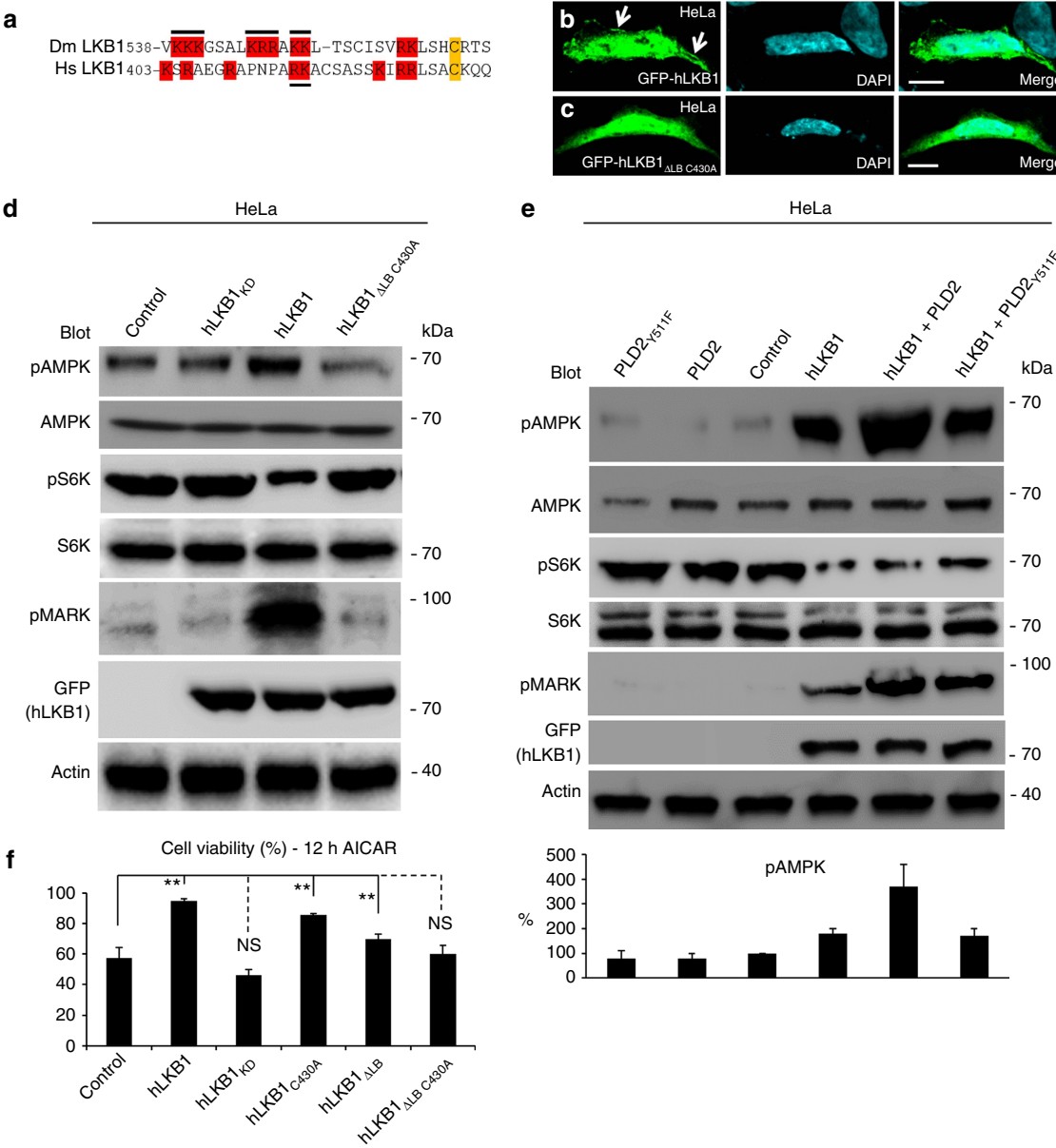

**Figure 4 | Membrane binding of hLKB1 is essential for its function in mammalian cells.** (**a**) Alignment of DmLKB1 and hLKB1 reveals several positively charged aa in the C terminus of hLKB1. Mutated residues are marked with bold lines. (**b,c**) GFP-hLKB1 localizes predominately to the nucleus and plasma membrane of HeLa cells cotransfected with STRADα (**b**), whereas GFP-hLKB1ΔLB C430A is not recruited to the cortex (**c**). (**d,e**) Immunoblottings of lysates from transfected HeLa cells, which lack substantial LKB1 expression, demonstrate an increase of activated AMPK (phospho-T172 AMPK), activated MARK (phospho-MARK) and decreased mTOR activation (measured by phosphorylation of S6K) upon GFP-hLKB1 + STRADα transfection, which is not observed in GFP-hLKB1ΔLB C430A + STRADα transfected cells (**d**). Co-transfection of PLD2 but not of a catalytically reduced variant (PLDY511F) together with hLKB1 results in a further increase of AMPK activation (**e**). The intensity of pAMPK bands was quantified (normalized against total AMPK) (**e**, lower panel, control was set as 100%). (**f**) Cell viability of HeLa cells under energetic stress (induced by incubation with AICAR for 12 h) was estimated using the MTT assay (as described in the methods section). Scale bars are 10 μm. Experiments were performed in triplicates. Error bars represent s.d. and statistical significance was determined using ANOVA: $P < 0.01$, **$P < 0.05$, not significant (NS).

hLKB1 strongly decreases its ability to activate AMPK in HeLa cells (Fig. 4d), which lack substantial endogenous LKB1 expression, and consequently leads to apoptosis in metabolically stressed cells (Fig. 4f). Apart from AMPK activation, hLKB1ΔLB C430A fails to activate two other kinases of the AMPK-family, MARK (Fig. 4d) and SadA (Supplementary Fig. 5a). Furthermore, hLKB1ΔLB C430A is no longer able to inhibit mTOR (as estimated by phosphorylation of p70-S6-Kinase 1, pS6K, Fig. 4d). Vice versa, addition of PA-enriched liposomes to a recombinant hLKB1/hSTRADα/hMo25 complex enhanced autophosphorylation of hLKB1 and

AMPK phosphorylation *in vitro*, whereas Phosphatidylcholine (PC) alone or PtdIns(4,5)P2 and PtdIns(3,4,5)P3-enriched liposomes had only a small effect on the activity of hLKB1 (Supplementary Fig. 3c,d).

**Overexpression of PLD2 enhances LKB1 activity.** PA is generated (among other pathways) from PC by two isoforms of PLD, PLD1 and 2 (ref. 38). Expression of PLD2 but not of a variant of PLD2 with a reduced catalytic activity (PLD2 Y511F (ref. 39))

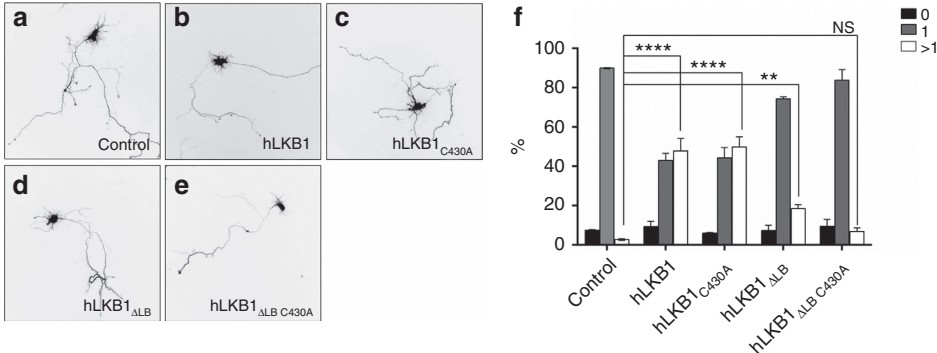

**Figure 5 | Induction of multiple axons in primary rat hippocampal neurons depends on membrane binding of hLKB1.** (**a–f**) Hippocampal neurons from E18 rat embryos were transfected at 0 d.i.v. with vectors for GFP (control), GFP-LKB1 or GFP fusion proteins for the indicated LKB1 mutants and STRAD and analysed at 3 d.i.v by staining with the Tau-1 antibody (axonal marker). Representative images of transfected neurons are shown. The scale bar is 20 μm. (**f**) The development of neuronal polarity was analysed by counting the number of neurons without an axon (0, black), with a single axon (1, gray) or with multiple axons (>1, white). Most primary rat hippocampal neurons transfected with GFP alone (control) develop only a single axon (**a**, quantified in **f**). In contrast, neurons overexpressing GFP-hLKB1 + STRADα or GFP-hLKB1$_{C430A}$ + STRADα (**b,c,f**) frequently establish two or more axons, whereas hLKB1 variants, which are deficient in membrane binding, fail to induce a multiple axon phenotype (**d–f**). Experiments were performed in triplicates. Error bars represent s.d. and statistical significance was determined using ANOVA: $P < 0.0001$, ****$P < 0.01$, **$P > 0.05$, not significant (NS).

together with hLKB1 results in a further increase of AMPK and MARK activation (Fig. 4e), suggesting that PLD2-induced production of PA enhances LKB1 activity *in vivo*. Conversely, inhibition of PLD reduces LKB1's capacity to activate AMPK (Supplementary Fig. 5b). However, basal levels of activated AMPK are maintained, indicating that other enzymes (e.g., Diacylglycerol kinase) can compensate loss of PLD activity. This is in line with the observation that *C. elegans*, *Drosophila* and mouse mutants lacking PLD enzymes do not exhibit as dramatic phenotypes as LKB1$_{ΔLB\ C564A}$ (refs 40–42).

**Membrane-binding-deficient LKB1 fails to induce multiple axons.** Overexpression of LKB1 induces the extension of multiple axons[43–45]. Consistently, overexpression of wild-type GFP-hLKB1 together with STRADα results in the formation of multiple axons by the majority of transfected rat hippocampal neurons in culture (Fig. 5b,f). Remarkably, expression of farnesylation-deficient hLKB1$_{C430A}$ induces this phenotype as efficiently as wild-type LKB1 (Fig. 5c,f). By contrast, mutation of the C-terminal basic region (hLKB1$_{ΔLB}$) almost abolished the effect on axon formation (Fig. 5d,f). Inactivation of the farnesylation signal in lipid-binding-deficient hLKB1 (hLKB1$_{ΔLB\ C430A}$) did result in further reduction of LKB1 activity in this assay (Fig. 5d,f). These results confirm the importance of the C-terminal membrane binding domain for the function of LKB1 in mammalian cells.

**Expression of LKB1 is downregulated in malignant melanoma.** As we have demonstrated that overexpression of PLD2 in cell culture results in increased activity of LKB1 (reflected by enhanced AMPK activation, Fig. 4e), we next assessed the pathophysiological relevance of our findings regarding tumour formation *in vivo*. Interestingly, expression of PLD2 is upregulated in several types of cancer, including melanoma, and high PLD2 expression correlates with poor survival rates of patients[46,47]. In cultured melanoma cells lacking detectable LKB1 expression (IGR37), membrane binding of LKB1 is essential for efficient AMPK (and MARK) activation, suppression of mTOR and cell survival under energetic stress (Supplementary Fig. 6).

Our data suggest that an aberrant increase in PLD expression in cancer cells will lead to increased PA levels and subsequent mTOR activation[48–50] only if in addition the expression of LKB1 is reduced, because PA-mediated LKB1/AMPK activation normally counteracts mTOR activity. To test this hypothesis, we investigated whether increased expression of PLD, aberrant activation of Akt and decreased LKB1 expression correlate with enhanced mTOR activity in melanoma. Indeed, biopsies of melanoma primary tumours show a strong correlation between these four parameters. In 81% of all analysed tumours PLD2 expression as well as phospho-Akt and mTOR activity were elevated while the expression of LKB1 was decreased (Fig. 6a–e). Only 12% of the samples showed mTOR activation in the presence of stimulating phospho-Akt and PLD2-overexpression although LKB1 staining appeared normal, which might be explained by inactivating mutations of LKB1, which do not affect protein expression or stability. In contrast, melanocytes in biopsies of healthy skin exhibited a strong LKB1 expression, as well as low level of PLD2, phospho-Akt and mTOR activation (Fig. 6a–d, arrows). Melanocytic nevi, a benign proliferation of melanocytes, which can give birth to malignant melanoma, show in many cases elevated PLD2 expression as well as activation of Akt, whereas LKB1 expression is still high in most cases, suppressing aberrant mTOR activation, which occurred only in 25% of the analysed specimen (Fig. 6a'–d'). Thus in melanoma, loss of LKB1 in PLD2 overexpressing tumours may further contribute to mTOR activation, whereas in benign neoplasia higher LKB1 level counterbalance aberrant PLD2 activation.

**Discussion**
In this study, we have elucidated a conserved mechanism regulating LKB1 activity and function *in vivo* and during tumour suppression (Fig. 7). We have further established the first link between PLD-mediated production of PA and activation of LKB1. Our data suggest that membrane targeting of LKB1 by direct binding to PA is essential for the function of LKB1 during *Drosophila* development and for activation of AMPK and suppression of mTOR activity in cultured mammalian cells. Furthermore, downregulation of LKB1 in malignant melanoma specimen, which exhibit activated Akt and overexpression of PLD2, correlates with enhanced mTOR activity, indicating that

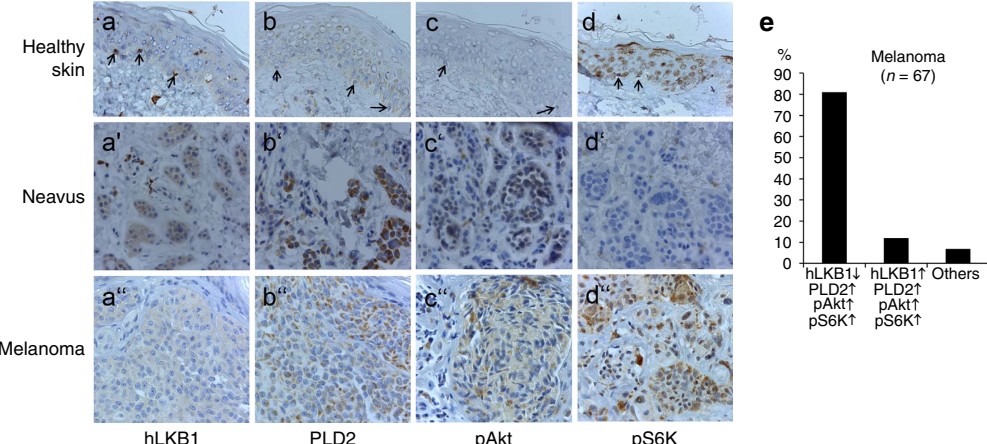

**Figure 6 | Downregulation of hLKB1 correlates with increased mTOR signalling in malignant melanoma overexpressing PLD2.** Immunostainings of paraffin-embedded tissue sections of normal skin (**a**–**d**), neavi (**a′**–**d′**) and primary malignant melanoma tumours (**a″**–**d″**). (**a**) hLKB1 is highly expressed in melanocytes in the *stratum basale* of the epidermis *in situ* (**a**, arrows), and still detectable at substantial levels (**a′**) whereas it is almost undetectable in the majority of malignant melanoma (**a″**). (**b**) PLD2 is not detectable in melanocytes of healthy skin sections (**b**, arrows) but becomes overexpressed in naevi (**b′**) and primary tumours (**b″**). (**c**) Akt is activated (indicated by staining of phospho Akt, pAkt) in naevi (**c′**) and malignant melanoma (**c″**), whereas it is not detectable in melanocytes of healthy skin samples (**c**). (**d**) Melanocytes in healthy skin biopsies (**d**) show only faint nuclear staining for phospho S6K (pS6K), which was used as to evaluate mTOR activation. Naevi (**d′**) exhibit only very few pS6K positive cells, whereas malignant melanoma show strong cytoplasmic and nuclear staining of pS6K (**d″**). (**e**) Quantification of melanoma biopsies scored for the indicated correlations. 'Others' are tumours, which do not exhibit increased phospho-Akt or PLD2 staining.

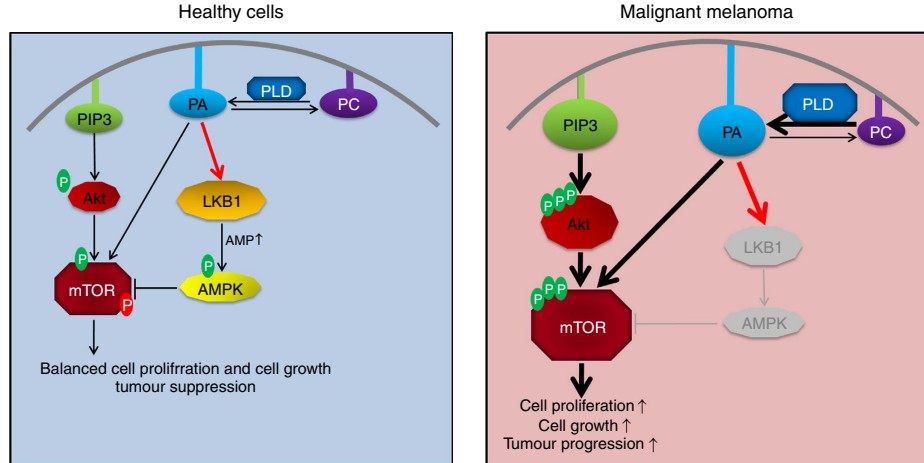

**Figure 7 | Simplified model of LKB1 activation by PLD2-produced PA and its downstream effects on AMPK activation and mTOR inhibition.** Under physiological conditions in non-tumourous cells, PA is produced by PLD (and other enzymes), resulting in an activation of mTOR and LKB1. Due to the inhibition of mTOR by LKB1-activated AMPK, the final result of increased PA level is a balanced cell proliferation and -growth. In contrast, in malignant melanoma, increased PI3K/Akt-signalling and enhanced production of PA by overexpression of PLD results in aberrant mTOR activation which is not counterbalanced by LKB1/AMPK and thus leads to increased cell proliferation and -growth and impaired tumour suppression.

the mechanism described in this study contributes to pathogenesis of malignant melanoma.

In contrast to previous studies investigating the role of LKB1 farnesylation in the oocyte of *Drosophila*[22], we found farnesylation of LKB1 to be dispensable for the correct protein localization in the embryonic epidermis and NBs and for the function of LKB1 in *Drosophila* development. This observation is in line with previous findings from cultured mammalian cells and mice. In malignant melanoma cells, farnesylation-deficient LKB1 suppresses colony formation at similar degree as the wild-type kinase[20]. Furthermore, studies from knock-in mice show that mice homozygous for farnesylation deficient LKB1 are viable and do not exhibit any obvious phenotypes[21]. However, in these mice AMPK is less phosphorylated, whereas in cell culture experiments, no difference was found between wild-type and farnesylation deficient LKB1 (ref. 51). Notably, several C-terminally truncated hLKB1 variants (due to nucleotide deletions or point mutations in STK11) with an intact kinase domain have been reported[52] and OMIM to be associated with Peutz-Jeghers syndrome, which might be due to the function of the C-terminal region in PA-mediated activation of LKB1.

Apart from the association of LKB1 with STRADα/Mo25, membrane binding and PA-mediated activation of LKB1 is a newly described upstream mechanism, which controls LKB1 kinase activity and function *in vitro* and *in vivo*. The regulation of LKB1 function by membrane recruitment raises the question whether this depends primarily on membrane localization or whether lipid binding itself stimulates LKB1 activity. Several

(phospho)-lipids have been demonstrated to activate or increase the activity of various serine/threonine kinases, including Phosphoinositide-dependent kinase 1 (PDK1, activated by PtdIns(3,4,5)P3 or PKCζ, Raf1, and mTOR (all activated by PA[49,53–55]). Fusion of GFP-LKB1$_{\Delta LB\ C564A}$ to lipid-binding domains (PH(PLCδ) and PH(Akt1)) resulted in a re-activation of GFP-LKB1$_{\Delta LB\ C564A}$ (Fig. 1o; Supplementary Fig. 4). Furthermore, the activity of LKB1$_{\Delta LB\ C564A}$ is strongly reduced in *in vitro* kinase assays (Fig. 3f,g). These findings could substantiate a hypothesis, in which the membrane functions as a scaffold for bringing together LKB1 and its cofactors (STRADα and Mo25) or its substrates. However, membrane binding deficient LKB1 robustly associates with its cofactors and one of its substrates (AMPK, Supplementary Fig. 3b). Moreover, addition of PA (but not of PtdIns(4,5)P2 or PtdIns(3,4,5)P3) is capable to substantially enhance the kinase activity of LKB1 *in vitro* (Fig. 3g; Supplementary Fig. 3c,d). Thus it is more likely, that in addition to membrane recruitment, PA functions to induce a conformational change of LKB1 thus enhancing its kinase activity. However we cannot exclude that in our *in vitro* kinase assays using immunoprecipiptated proteins from transfected cells wild-type LKB1 might have co-immunoprecipitated with PA-enriched micelles, which enhanced its kinase activity in this experiment.

Interestingly, PLD-produced PA activates both, LKB1 (this study) and mTOR[48–50,55]. Up to now, overexpression of PLD and increased levels of PA have been only assigned to pro-oncogenic pathways, in particular mTOR and Raf signalling[54–58]. Our data thus demonstrate the first anti-oncogenic mechanism of PLD function, which might counterbalance its pro-oncogenic effects as long as the LKB1-AMPK signalling axis is intact (Fig. 7).

Notably, Mukhopadhyay *et al.* showed recently, that in a mesenchymal-like breast cancer cell line (MDA-MB-231), which expresses robust levels of endogenous LKB1 (in contrast to the majority of breast cancer metastases), activated PLD induces a reduction in phospho-AMPK[59]. Thus, PLD-mediated activation of LKB1 might be either cell type specific or depends on the basal activation of both signalling cascades, the LKB1-AMPK axis and the PI3K-AKT pathway.

The LKB1-AMPK signalling pathway is well described to play a crucial role for cell polarity, energy homeostasis and cell proliferation and is frequently disturbed in various types of cancer[16,17]. As mentioned above, one particular function of AMPK is to counterbalance the pro-proliferative activity of mTOR[8–15]. Defects in the LKB1-AMPK signalling pathway (resulting in impaired inhibition of mTOR) have been already demonstrated to contribute to BRAF- and KRAS-induced melanoma formation in mouse models[60–63]. Remarkably, somatic mutations in human STK11 (encoding LKB1) are rather rare in malignant melanoma[64,65]. However, our data demonstrate that downregulation of LKB1 protein expression can be observed in a majority of malignant melanoma samples and therefore most likely contributes to the pathogenesis of this type of skin cancer.

Our study provides strong evidences for an important and conserved regulatory mechanism, which facilitates membrane recruitment of LKB1 by PA and is essential for the kinase activity and function of LKB1 during development and tumour suppression. The dissection of upstream regulatory mechanisms of LKB1 contributes to a better understanding of the physiological regulation of LKB1 and their misregulation in tumours, which might be useful for the development of therapeutic approaches in the future.

## Methods

**Fly stocks and genetics.** UASt::GFP-LKB1 and *lkb1*::GFP-LKB1 transgenes were generated using phiC31-mediated germ line transformation on attp40. For rescue experiments of different LKB1 variants, we used the *lkb1^x5* null allele. Lethality tests were performed in three independent experiments with $n = 100$ in each experiment. For rescue experiments using the UAS/GAL4 system, we used UAS::GFP-LKB1 and actin5C::GAL4 instead of *lkb1*::GFP-LKB1.

**DNA and constructs.** Cloning of the cDNA of wild-type LKB1 into pENTR was performed using standard PCR on a full length EST clone (*Drosophila* Genomics Resources Center, DGRC) as template using the following primers: LKB1-F: 5′- CACCATGCAATGTTCTAGCTCTCGG-3′, LKB1-R: 5′-CTACGAAGTTCG-GCAGTGG-3′. Similar, truncated fragments of LKB1 were cloned with the following oligonucleotides: LKB1$_{512}$-F: 5′-CACCATGCACACCTACGAACCGCC-3′, LKB1$_{536}$-F: 5′-CACCATGGCGCCCGTCAAGAAG-3′, LKB1$_{552}$ -F: 5′-CACCAT-GCTGACGTCCTGCATCTCCG-3′. For expression of LKB1 from its endogenous promoter we inserted a genomic fragment (from 2.8 kbp upstream of the translation start to 1 kbp downstream of the stop codon) into pENTR using the following primers: LKB1$_{gen}$-F: 5′- CACC CACTAGCGTAATTTGACGG-3′, LKB1$_{gen}$-R: 5′- CTC GAG CAGCAGTACGGTCATCTC-3′. An XbaI-cutting site was introduced replacing the start codon using mutagenesis PCR and the following primer: LKB1$_{gen}$-XbaI-F: 5′-GGCTCCGCGGAGGTTTTCTAGACAATGTTCTA-GCTCTC-3′. Subsequently, GFP was inserted into the XbaI site by PCR and standard ligation. Mutagenesis PCR was used to generate defined point mutations with full length or genomic LKB1 cDNA in pENTR as template. The following oligonucleotides were used for mutagenesis (mutation underlined): LKB1$_{\Delta LB}$: Combination of 1. LKB1$_{K546A\ R547A\ R548A\ K550A\ K551A}$-F: 5′-TCGGCACTG-GCGGCGGCCGCCGCGGCGGCTGACGTCCTGC-3′ and 2.LKB1$_{K539A\ K540A\ K541A}$-F: 5′-GAGGAGGCGGCCCGTCGCCGCGGCGGGATCGGCACTG-3′, LKB1$_{C564A}$-F: 5′-GTGCGCAAGCTTAGCCACGCCCGAACTTCGTAG, LKB1$_{D317A}$-F: 5′- CAAACGCTGAAGATTTCCGCCTTCGGTGTGGCG.

To express LKB1$_{\Delta LBC564A\ PH(PLD)}$ and LKB1$_{\Delta LBC564A\ PH(Akt)}$, the PH domain of PLCδ and Akt was amplified by PCR and ligated via an endogenous Bpu1101-I site into lkb1::LKB1 pENTR using the following primers: PH(PLCδ)-F: 5′- GCTG-AGCCACGCCCGAACT TCGgatgaggatctacaggcgct-3′, PH(PLCδ)-R: 5′- GCTGA-GCTAGATCTTGTGCAGCCCCAG-3′, PH(Akt)-F: 5′-GCTGAGCCACGCCCG-AACTTCGGTCGTAAAGGAGGGGTGG-3′ and PH(Akt)-R: 5′- GCTGAGCTT-ATATGAGCCGGCTGGATAC-3′.

For expression of hLKB1, the open reading frame of hLKB1 was cloned into pENTR using the following nucleotides: hLKB1-F: 5′- CACC ATGGAGGTG-GTGGACCC-3′ and hLKB1-R: 5′- TCACTGCTGCTTGCAGG-3′. For mutation of the farnesylation, lipid binding motif and construction of the kinase dead version, the following nucleotides were used in mutagenesis PCRs: hLKB1$_{C430A}$-F: 5′- CGCCGGCTGTCGGCCGCTAAGCAGCAGTGAAAGGGT-3′, hLKB1$_{R415AK416A}$-F: 5′- GCCCCCAACCCTGCCGCCGCGGCCTGCTCCGCC-AGC-3′ and hLKB1$_{D194A}$ 5′- ACCCTCAAAATCTCCGCCCTTGGCGTGGCCG-AGGCA-3′.

Constructs were recloned into GFP-tagged destination vectors containing a One-Strep-Tag fused to the N terminus of GFP (USGW, modified TGW, Murphy lab, DGRC, expression in S2R cells) or into a modified EGFP-C1 vector (CGW, expression in mammalian cells) containing a gateway cassette using the gateway technology (Life technology).

**Antibodies.** Antisera directed against full length LKB1 were raised by injection of a fusion protein of LKB1 and MBP into two guinea pigs (Amsbio, Abingdon, UK).

**Cell culture and cell viability assay.** HeLa, IMR90, IGR37 and MDCK cells were obtained from ATCC. All cells were maintained in Dulbecco's modified Eagle's medium supplemented with 10% fetal calf serum, 2 mM Glutamine at 37 °C in a 5% $CO_2$ atmosphere and negatively tested for mycoplasma contamination by PCR. Cells were transfected using FUGENE (Promega) according to the manufacturer's instructions. For evaluation of cell viability upon energetic stress, $50 \times 10^3$ cells/well were transiently transfected in a 24well plate with GFP-hLKB1 + STRADα constructs and treated for 12 h with 2.5 mM AICAR (Santa Cruz Inc.). GFP + STRADα were used as negative control. Cell viability was assessed in triplicates using MTT assay according to the manufacturer's instructions (SIGMA).

**Western blotting and coimmunoprecipitation.** Western blotting was done as previously described[66]. For mammalian cell culture experiments, HeLa or IGR37 cells were transiently transfected with the indicated constructs. 48 hours after transfection, cells were incubated with 2 mM AICAR for 1 h and harvested in lysis buffer (1% Triton X-100, 150 mM NaCl, 1 mM $CaCl_2$, 1 mM $MgCl_2$, 50 mM TRIS-HCl pH 7.5) supplemented with protease- and phosphatase inhibitors. For embryonic lysates, *lkb1*::GFP-LKB1 expressing embryos were collected from overnight plates. Coimmunoprecipitation using GFP-Trap (ChromoTek) of GFP-LKB1 with HA-tagged DmSTRADα (Stlk) and myc-tagged DmMo25 was done in embryonic lysates, which ubiquitously expressed the proteins using arm::GAL4. Primary antibodies used for western blotting were as follows: rabbit anti-Actin (1:1,000, sc-47778, Santa Cruz), mouse anti-S6K (1:500, sc-8418, Santa Cruz ), rabbit phosphoT389-S6K (1:500, sc-11759, Santa Cruz), rabbit anti-phospho-T172-AMPK (1:200, sc-33524, Santa Cruz), rabbit anti-phospho-MARK1/2/3 (1:500, PA5-17495, Thermo Scientific), mouse anti-myc (1:100, 9E10, DSHB), rabbit anti-AMPK (1:400, sc-25792, Santa Cruz), guinea pig anti LKB1

(1:500, this study), mouse anti-GFP (1:500, sc-9996, Santa Cruz), mouse anti-HA (1:500, #11583816001, Roche), rabbit anti-GST (1:5,000, #G7781, SIGMA). For statistical analysis, three independent experiments were scored. Intensity of the bands was quantified by ImageJ.

**Immunohistochemistry.** *Drosophila* embryos were fixed in 4% formaldehyde, phosphate buffer pH 7.4 as described before[67]. Primary antibodies used for indirect immunofluorescence were as follows: guinea pig anti LKB1 (1:250, this study), rabbit anti Baz (1:2,000, kindly provided by A. Wodarz), mouse anti α-spectrin (3A9, 1:50, DSHB), mouse anti Dlg (4F10, 1:50, DSHB), rat anti DE-Cad (DCAD2, 1:25, DSHB), rabbit anti GFP (1:400, sc-8334, Santa Cruz Inc), guinea-pig anti Miranda (1:1,000, kindly provided by A. Wodarz). HeLa cells were fixed with 4% PFA in PBS and stained with the rabbit anti-phospho-Sad antibody (1:250) and a mouse anti-LKB1 antibody (1:200, sc-32245, Santa Cruz) in 10% goat serum.

Secondary antibodies conjugated with Alexa 488, Alexa 568 and Alexa 647 (Life Technology) were used at 1:400. Images were taken on a Zeiss LSM 710 Meta confocal microscope and processed using Adobe Photoshop.

For immunohistochemical staining of healthy skin, nevi and melanoma, a tissue micro array (TMA) of paraffin-embedded healthy skin, nevi and melanoma primary tumours was analysed. Paraffin sections were deparaffinized for 30 min at 72 °C, washed two times for 7 min in Xylol and subsequently re-watered in a descending sequence of ethanol/water mixture. Prior to staining sections were subjected for 5 min to heat-induced epitope-retrieval (HIER) using 1 mM Tris-EDTA-buffer (pH 8.5) at 120 °C. Sections were blocked in peroxidase-blocking solution (Dako, #S2023) for 5 min at room-temperature and washed 5 min with wash buffer (Dako, #S3006) prior to incubation with primary antibody diluted in antibody diluent (Dako, #S2022) for 30 min. Primary antibodies were as follows: rabbit anti-LKB1 (D60C5F10, 1:200, Cell Signaling), rabbit anti-phospho-Akt (1:20, Cell Signaling #4060), rabbit anti-PLD2 (1:1,000, Cell Signaling #13891) and rabbit anti-S6K-phospho-T389 S6K (1:50, Cell Signaling #9206). Subsequently, sections were washed with wash buffer and incubated for 30 min with HRP-coupled secondary antibody (Dako EnVision, # K5007). Stainings were developed after washing using DAB/chromogen solution (Dako, #K5007). To visualize cellular structures, stained sections were counterstained with hematoxylin (Merk, #10517505000) for 1 min and dehydrated in ethanol/xylol before embedding. The staining intensity was determined blinded for all tissue samples as followed: negative-0, weak-+, moderate-++ and strong-+++. LKB1 exhibited a strong expression in melanocytes *in situ*, so weak or negative intensity was scored as downregulation of the protein, whereas moderate staining was classified as slightly downregulated. Activated Akt was negative in healthy skin biopsies, so weak, moderate and strong staining was classified as upregulated. PLD2 expression was negative or weak in melanocytes *in situ*, thus moderate and strong staining was scored as upregulated. Phospho-S6K was strongly expressed in the nucleus but not in the cytoplasm of melanocytes but cytoplasmic staining occurred only in malignant tumours. Consequently, weak to strong cytoplasmic staining of phospho-S6K was classified as upregulated.

**Lipid binding assays.** Fusion proteins of the C terminus of LKB1 (aa 353–567) with MBP were expressed in E.coli and affinity purified. Lipid strips (Echelon) were incubated over night with purified MBP-LKB1 fusion proteins at 0.5 µg ml$^{-1}$ in TBST containing 3% BSA, washed and probed with antibodies against MBP (1:1,000, Santa Cruz sc-73416) as described above.

For membrane floatation and *in vitro* kinase assays, lipids were obtained from Avanti Polar Lipids (Egg-PA, #840101, Egg-PC #840051, Brain PtdIns(4,5)P2 #840046) and Echelon (PtdIns(3,4,5)P3 #P-3916). Liposomes (10 mM total lipid concentration, either PC alone or PC:PA/PtdIns(3,4,5)P3/PtdIns(4,5)P2 in a 9:1 molar ratio) were prepared in LB buffer (30 mM Tris, 4 mM EGTA, pH 8.0) by extrusion through a 0.1 µm polycarbonate membrane using a Mini-Extruder (Avanti Polar Lipids). The membrane floatation was performed as follows[68]: liposomes (100 µl of 10 mM total lipid concentration) were incubated on ice for 30 min with 1 µg recombinant protein. LB buffer (30 mM Tris, 4 mM EGTA, 2 M sucrose (pH 8.0)) was added to the incubation reaction to bring the final sucrose concentration to 1.6 M, and this mixture was overlaid with cushions containing 1.4 M, 0.4 M, and 0.25 M sucrose in the same buffer in a TLA-55 tube. After centrifugation at 186,000g (4 °C) for 45 min in a TLA-55 rotor (Beckman), the 0.25/0.4 M interphase (top fraction, T) and the loading fraction (bottom fraction, B) were collected and analysed by SDS-PAGE and western blot. For statistical analysis, three independent experiments were scored. Intensity of the bands was quantified by ImageJ.

***In vitro* kinase assay.** OneStrep-GFP-LKB1 plus OneStrep-GFP-Stlk were precipitated from transfected S2R cells (2 mg total protein lysates) using Streptactin beads (IBA, Goettingen, Germany). The beads were washed five times in harsh washing buffer (50 mM TRIS pH 7.5, 500 mM NaCl) and one time in LKB1 kinase buffer (50 mM TRIS pH 7.5, 10 mM MgCl$_2$, 10 mM MnCl$_2$, 1 mM DTT, 100 µM ATP, phosphatase- and protease inhibitors). Immunoprecipitated proteins were then incubated with 2 µg of recombinant GST-DmAMPKα$_{108-280}$ (= aa 108–280, containing the T-loop with the LKB1-phosphorylation site) and

0.3 µCi[$_{γ-32}$ATP] in kinase buffer for 1 h at 30 °C. The reaction was terminated by addition of SDS sample buffer and samples were subjected to SDS-PAGE. Phosphorylation was detected by exposure to X-ray films and quantified with Aida 2D Densitometry software.

For assays with human recombinant kinase complex (Supplementary Fig. 3b,c), 1 µg of a complex of recombinant hLKB1, STRADα and Mo25 (SIGMA) was used as described above replacing immunoprecipitated LKB1 protein. In this experiment, GFP-AMPKα1 from transfected HeLa cells (2 mg total protein lysate) was used as substrate as a recombinant fragment would have interfered with the autophosphorylation bands. After isolation of precipitated OneStrep-GFP-AMPK, recombinant hLKB1/STRADα/Mo25 was added and incubated as described above. Subsequently, OneStrep-GFP-AMPK was purified from the reaction mixture using Streptactin beads.

For kinase assays with addition of lipids, Liposomes were prepared as described above and added at 10 mM final concentration.

For statistical analysis, three independent experiments were scored. Intensity of the bands was quantified by ImageJ.

**Transfection and analysis of neurons.** Hippocampal neurons were isolated from the brains of E18 rat embryos and cultured as described previously[69]. Dissociated hippocampal neurons were plated at 70,000 cells per well in a 24 well plate and transfected 3 h after plating by calcium phosphate co-precipitation. Neurons were fixed at 3 DIV and permeabilized with 0.1% Triton X-100, 0.1% sodium citrate in PBS for 3 min on ice. Neurons were stained with the Tau-1 antibody as axonal marker (Chemicon, MAB3420; 1:300) and Alexa-Fluor-conjugated secondary antibodies (Molecular Probes; 1:300). The stage of neuronal differentiation and axon formation was determined according to published criteria[70]. For each LKB1 variant, 3 different experiments with 100 analysed neurons in each were scored.

**Statistics.** All experiments were performed in triplicates. Error bars represent s.d. and statistical significance was determined using ANOVA: $P < 0.0001$, ****$P < 0.001$, ***$P < 0.01$, **$P < 0.05$, *$P > 0.05$, not significant (NS).

**Data availability.** The authors declare that all data are available within the Article and Supplementary Files, or available from the authors upon request.

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

## Acknowledgements

We thank A. Wodarz, B. Schmidt, G. Längst, the Bloomington *Drosophila* stock center at the University of Indiana and the Developmental Studies Hybridoma Bank at the University of Iowa for providing reagents. This work was supported by grants of the DFG to M.P.K. (DFG3901/2-1, DFG3901/1-2, SFB699/A13) and to A.W.P. (DFG PU 102 12-1 and Cells-in-Motion Cluster of Excellence (EXC 1003—CiM)).

## Author contributions

G.D., L.K., C.T., P.D., O.P. and G.M. designed and performed the experiments, A.W.P. and M.P.K. designed and directed the project and wrote the manuscript. All authors discussed results and commented on the manuscript.

## Additional information

**Competing interests:** The authors declare no competing financial interests.

