## [Peer Review File · Nature Communications]

Reviewers' comments:

Reviewer #1 (Remarks to the Author): Expert in PLD and cancer

The manuscript "Membrane binding of LKB1 and its activation by phosphatidic acid is essential for its function in development and tumour suppression" builds on the interesting finding that LKB1 has a polybasic sequence near its carboxy terminus that appears to mediate membrane localization through binding to anionic lipids and preferentially phosphatidic acid (PA). There are some intriguing findings in the paper; nonetheless, methodological weaknesses related to current standards for data reproducibility and some areas of overinterpretation lend a sense of prematurity to the work.

Specific points

1) Bottom of page 2, text; Fig 1o. A major criticism of the manuscript is the substandard statistical analysis. Using Fig. 1o as the initial example, error bars are shown, but it is unknown what they represent. The legend to Fig. 1 states "n=300 for each genotype" - but the number of flies examined in a singled experience can not in itself be used to generate an error bar, nor is it clear what the error is showing - e.g. STDEV, 2 STDEVs, or SEM - and no significance is indicated.

2) Fig. 2A - "a larger C-terminal fragment encompassing the farnesylation motif and a polybasic stretch of amino acids within the C-terminus of DmLKB1" - note that the poly basic region starts at aa 538, but this fragment started at 512, i.e. an extra 26 aa. While this might not seem like something worth commenting on, the authors should take a look at Mike Housley's paper (PMID:11994273) on defining a PA-binding motif in phosphodiesterase (PDE4A1); these authors found that there was a protein-interaction motif immediately adjacent to the PA-binding polybasic region that was important in the membrane localization, so a little more accuracy in description here would be advisable.

3) Fig. 2B - lipid-binding westerns "PIP-strips" are just an initial method for demonstrating binding, and subject to many forms of false positive and negative results, as well as not being very quantitative. The authors should perform a vesicle sedimentation experiment (i.e., liposomes containing or not containing the lipids of interest in varied amounts) to validate the PIP-strip findings. In particular, PI5P binding is also lost, and PIP3 to a lesser extent - not just PA.

4) Context - this is not the initial report of a polybasic motif binding to PA... e.g. PMID:19325080, PMID:19345277 - there are probably a dozen or so by now. Citing some of these would provide context for the current finding. In general, the literature is quite poorly cited with respect to PLD and PA studies.

5) Fig. 3D - recovering stability, localization, and function with addition of the PH domain of PLD (which PLD?) or AKT seems like an odd experiment to do, since these domains don't bind PA. In Sup. Fig 2, the addition of the AKT PH domain seems to mediate membrane localization - but in fact, the construct with the PLD PH domain appears to be cytosolic - there is no obvious co-localization with spectrin.

6) Page 4, "Finally, addition of PA to recombinant hLKB1-STRAD α -Mo25 increases the kinase activity of the LKB1 complex by 4fold (Fig. 3g)." First, the graph doesn't show this - it looks like a 2.5-fold increase. Second, there are error bars of unknown definition, but no mention of the number of replicates in the experiment nor how many times the experiment was performed... Fig. 3f is described as n=3, but otherwise similarly poorly qualified. How was experiment 3g performed? What does "addition of PA" mean? Short-chain soluble PA? PA liposomes? PC liposomes containing some amount of PA? What about PI5P, the other lipid from Fig. 2B that binds to the polybasic sequence? How were the blots quantified?

7) Fig. 4b - what do the asterisks signify? How were the statistical calculations perform? Multiple t-tests? Anova? Why are only some combinations examined? Were the experiments scored in a blinded fashion?

8) Page 5, "Expression of PLD2 together with hLKB1 results in a further increase of AMPK activation (Fig. 4e), suggesting that PLD2-induced production of PA enhances LKB1 activity in vivo." - This is an interesting preliminary finding, but is nothing more than suggestive without

further controls and extension of the work. What does PLD2 do by itself in this assay? What about transfecting a catalytically-inactive PLD2 allele? Does LKB1 require PLD activity to activate AMPK? There are very good PLD inhibitors developed by the Frohman and Brown groups that are commercially and inexpensively available from Sigma and other companies. Perhaps the PA is instead being generated by a DAG kinase? If LKB1 deficiency is embryonic lethal, why are mice lacking both PLD1 and PLD2 viable and relatively normal?

9) Fig. 4g-k - interesting, but how was the quantitation done? Was it scored in a blinded fashion? There appears to be a lot of pS6 in the normal tissue as well. Finally, this sort of experiment is only correlative...

10) In the absence of better validation of the role of PLD (e.g. loss-of-function experiments), it is not possible to conclude (page 6) that the authors "have established here the first link between PLD-mediated production of PA and activation of the tumour suppressor LKB1." The PA, if important, could just as easily come from DAG kinase isoforms, or even PLD1. Showing that something happens with overexpression does not prove that that linkage occurs with endogenous proteins. For an example, see Y. Kanaho's work, PMID:23109426.

11) Is there not supposed to be a materials and methods section in the paper? I could not find one. It is important to provide sources of key plasmids, antibodies, etc., as well as the methods that would be needed for others to attempt to reproduce the work.

Reviewer #2 (Remarks to the Author): Expert in drosophila neurodevelopment and cancer

In this manuscript, the authors attempt to demonstrate that the phospholipid binding by LKB1 is an upstream regulatory mechanism essential for LKB1 activity. The authors first showed that apart from its nuclear localizing inactive form, LKB1 could also localized to the cell membrane in mammalian and Drosophila epithelial cells as well as Drosophila embryonic neuroblasts. They discovered that the C-terminal polybasic stretch of amino acids is essential for its membrane localization particularly through interacting with phosphatidic acid. In addition, it was observed that the membrane binding deficient form of LKB1 exhibits impaired functionality in rescuing embryonic lethality and disturbed anterior-posterior polarity in Drosophila oocyte and axon formation in primary hippocampal neurons. Through kinase assay, the authors further showed that GFP-LKB1 Δ LB C564A displayed impaired kinase function, in particular with reduced AMPK phosphorylation. As a result, AMPK-mediated inhibition on S6K phosphorylation and thus mTOR activation was reduced. LKB1-mediated cell viability was reduced when GFP-LKB1 Δ LB C564A was used. Interestingly, the authors showed that co-expression of PLD2 with LKB1 results in significant enhancement of LKB1-mediated phosphorylation of AMPK, which in turn leads to suppression of mTOR pathway. Lastly, the authors established a clinical relevance for their findings with human Melanoma, by demonstrating that a huge proportion of human Melanoma tumor exhibit reduced LKB1 level with increased PLD2 expression and Akt and mTOR activity. In a nutshell, this is a very novel and nice study and the authors have convincingly provided the first evidence on the membrane phospholipids binding domain of LKB1 and its importance for regulating LKB1 functionality. Furthermore, they have also provided evidence on the potential interaction between LKB1 and PLD, which is up-regulated in several cancer types. This manuscript will be of broad interest to readers of Nature Communications. I recommend publication of this manuscript provided that the authors are able to address the following concerns of the reviewer.

Major comments:

Mukhopadhyay et al 2015 demonstrated a reciprocal regulation of AMPK and PLD that increase in PLD activity reduced AMPK activity, while increase of AMPK activity reduced PLD activity. This paper needs to be cited. Mukhopadhyay et al 2015 showed that AICAR treatment increased p-AMPK, which is suppressed by addition of PA. However, authors of this manuscript showed that LKB1-induced phosphorylation of AMPK is enhanced by PLD2 expression. The authors need to explain the reason for these conflicting conclusions and probably provide additional evidence to support that PLD2 functions upstream to enhance the activity of LKB1.

Minor comments:

1. "However, overexpression of GFP-DmLKB1 Δ LB C564A in *lkb1*-mutant flies does not result in detectable rescue capacity (data not shown).

This data is already shown in Fig. 1o: the lethality/survival chart.

2. "... or kinase-dead hLKB are inactive and fail to induce multiple axons (Fig. 4e)"

The kinase-dead hLKB1 data was not found in Fig 4.

3. Fig 2H lacks a wt control.

4. Colour scheme in Fig 1o can be improved.

5. In Fig 3d, GFP-LKB1 Δ LB C564A was almost undetectable, inconsistent with fig3c expression. This extremely low levels of protein will interfere with the conclusion for rescue.

6. Fig 3g quantification does not correlate with kinase assay data.

Reviewer #3 (Remarks to the Author): Expert in AMPK signalling

A tumor suppressor protein kinase LKB1 plays diverse roles in various biological processes, including cell polarity, growth and also metabolism by regulating AMP-activated protein kinase (AMPK) and 12 other related kinases. The current manuscript claims that LKB1 is targeted to membrane through direct binding with phospholipids, which plays important role in fully activating AMPK and induction of multiple axons in primary neurons. However, there are several controls and descriptions of key experiments lacking and thus the authors claims are not compellingly supported by robust sets of data.

Major comments:

1. First of all, I could not find method section in the manuscript. Therefore, I am not able to judge if the experiments were done using standard/appropriate protocols. Moreover, there is no information regarding sample numbers (n), replicates, and descriptions of statistical analysis in the legend.

2. The authors have no comment (discussion) and measurement of AMPK-related kinases in this study. These kinases (e.g. SAD/BRISK, MARKs) have been shown in several studies that they play important roles in cell polarity and axon induction. It is necessary to show if at least some of those kinases is activated more strongly upon membrane localization of LKB1.

3. It is unclear the reason why membrane localization of LKB1 more robustly activate AMPK. Is it because AMPK and LKB1 co-localize at the membrane (as proposed by Bruce Kemp and Dario Alessi groups) or intrinsic activity of LKB1 is increased upon binding to phospholipids? Concerning the latter, measurement of endogenous LKB1 activity (IP kinase assay) should clarify this point.

4. The authors used several point and truncation mutants of LKB1 in this study. It is necessary to show control experiments that such mutations/truncations do not alter intrinsic activity and/or binding to STRAD/MO25. There is no data showing when mutants were introduced to flies/cells, if

they form function/stoichiometric complex with STRAD/MO25.

5. Fig 3f, the authors describe that they measure LKB1 activity in vitro using STRAD as substrate. This is not a standard assay (no rationale) and the authors should show absolute activity (^{32}P incorporation into the substrate (STRAD)/min/mg). Fig4e (right panel), the blots are not publication quality and higher pAMPK signal appears to be simply due to higher loading (based on total AMPK blot).

Response to the reviewers

Reviewer #1 (Remarks to the Author): Expert in PLD and cancer

The manuscript "Membrane binding of LKB1 and its activation by phosphatidic acid is essential for its function in development and tumour suppression" builds on the interesting finding that LKB1 has a polybasic sequence near its carboxy terminus that appears to mediate membrane localization through binding to anionic lipids and preferentially phosphatidic acid (PA). There are some intriguing findings in the paper; nonetheless, methodological weaknesses related to current standards for data reproducibility and some areas of overinterpretation lend a sense of prematurity to the work.

Specific points

1) Bottom of page 2, text; Fig 1o. A major criticism of the manuscript is the substandard statistical analysis. Using Fig. 1o as the initial example, error bars are shown, but it is unknown what they represent. The legend to Fig. 1 states "n=300 for each genotype" - but the number of flies examined in a singled experience can not in itself be used to generate an error bar, nor is it clear what the error is showing - e.g. STDEV, 2 STDEVs, or SEM - and no significance is indicated.

- We are sorry for the error that led to the missing information about the statistical analysis. The statistics were explained in the Methods section, which was unfortunately lost during the transfer process. All lethality tests were averaged on three independent experiments, error bars indicate STDEV and significance was calculated using ANOVA:

"Statistics

All experiments were performed in triplicates. Error bars represent standard deviation and statistical significance was determined using ANOVA. $p < 0.0001$ **, $p < 0.001$ ***, $p < 0.01$ **, $p < 0.05$ *, $p > 0.05$ not significant (ns)."**

In Fig. 1o (lethality tests), we indicated the significance only for surviving flies to make the figure easier to understand. In this particular case, significance might even be misleading: For instance the rescue efficiency (as estimated by the amount of survivors) between wild type GFP-LKB1 and GFP-LKB1 Δ LB C564A PH(Akt) is significant (), but the fact that GFP-LKB1 Δ LB C564A PH(Akt) can rescue at all (in contrast to GFP-LKB1 Δ LB C564A) is the most important result in this figure.*

2) Fig. 2A - "a larger C-terminal fragment encompassing the farnesylation motif and a polybasic stretch of amino acids within the C-terminus of DmLKB1" - note that the poly basic region starts at aa 538, but this fragment started at 512, i.e. an extra 26 aa. While this might not seem like something worth commenting on, the authors should take a look at Mike Housley's paper (PMID:11994273) on defining a PA-binding motif in phosphodiesterase (PDE4A1); these authors found that there was a protein-interaction motif immediately adjacent to the PA-binding polybasic region that was important in the membrane localization, so a little more accuracy in description here would be advisable.

- Originally we added more amino acids in addition to the polybasic motif to demonstrate that the C-terminus is important. We agree that this sequence might contain an up to now unidentified domain/motif although no reported (or predicted) domains are present between the kinase domain and the polybasic motif. Thus we repeated the experiment with only aa

536-567 fused to GFP (Fig. S2a) and detected a similar membrane localization of the chimeric protein in S2R cells.

3) Fig. 2B - lipid-binding westerns "PIP-strips" are just an initial method for demonstrating binding, and subject to many forms of false positive and negative results, as well as not being very quantitative. The authors should perform a vesicle sedimentation experiment (i.e., liposomes containing or not containing the lipids of interest in varied amounts) to validate the PIP-strip findings. In particular, PI5P binding is also lost, and PIP3 to a lesser extent - not just PA.

- We agree and added a new experiment (Fig. S2b) using a liposome flotation assay which clearly shows that the C-terminal polybasic motif in LKB1 mediates strong binding to PA (but only weaker to PIP2 and PIP3). We did not include PI5P as this lipid is predominately enriched in nuclear and endoplasmic reticulum membranes (Jones et al. 2006) where LKB1 is not found at substantial amounts in vivo.

4) Context - this is not the initial report of a polybasic motif binding to PA... e.g. PMID:19325080, PMID:19345277 - there are probably a dozen or so by now. Citing some of these would provide context for the current finding. In general, the literature is quite poorly cited with respect to PLD and PA studies.

- We are happy to cite now more PA-interacting proteins and PLD- references – the first version of our manuscript was originally restricted to 30 references (transfer from NCB).

5) Fig. 3D - recovering stability, localization, and function with addition of the PH domain of PLD (which PLD?) or AKT seems like an odd experiment to do, since these domains don't bind PA. In Sup. Fig 2, the addition of the AKT PH domain seems to mediate membrane localization - but in fact, the construct with the PLD PH domain appears to be cytosolic - there is no obvious co-localization with spectrin.

- We apologize for the confusing labelling of PH(PLD), which is the PH-domain from Phospholipase C δ .

- Indeed the staining in the original figure was rather weak. To improve the quality of the figure we included a new figure for LKB1 Δ LB CA-PH (PLC δ) (now Fig. S4). We furthermore changed the colors in this figure (green for LKB1-chimera, red for spectrin) to better visualize the colocalization of LKB1 Δ LB CA-PH (Akt/PLC δ) and spectrin.

- The PH domains were used to test the possibility that membrane localization is required for the function of LKB1 in vivo, which is clearly shown by our results. As we show in Fig. 3, PA stimulates the kinase activity LKB1. It is possible that the replacement of the PA-binding polybasic stretch by the PH domains not only confers membrane localization to the artificial fusion protein but also PIP2 or PIP3 regulation. However, given the facts that binding to PIP2 or PIP3-enriched liposomes is rather weak (Fig. S2b) and that addition of PIP2 and PIP3 does not (or only slightly) enhance the kinase activity of LKB1 in vivo (Fig. 3g and S3b), we think that there is a simple explanation: The PH-domain of PLC δ is not absolutely specific for PIP2 but also binds robustly to PA-containing liposomes (Pawelczyk et al. 1999, PMID 10336610). Similar, Akt1 is capable to bind PIP3 and PA (Mahajan et al. 2010, PMID 20333297 and Bruntz et al., PMID 24257753). We discuss this point now in the revised manuscript.

6) Page 4, "Finally, addition of PA to recombinant hLKB1-STRAD α -Mo25 increases the kinase activity of the LKB1 complex by 4fold (Fig. 3g)." First, the graph doesn't show this - it looks like a 2.5-fold increase. Second, there are error bars of unknown definition, but no mention of the number of replicates in the experiment nor how many times the experiment was performed... Fig. 3f is described as n=3, but otherwise similarly poorly qualified. How was experiment 3g performed? What does "addition of PA" mean? Short-chain soluble PA? PA liposomes? PC liposomes containing some amount of PA? What about PI5P, the other lipid from Fig. 2B that binds to the polybasic sequence? How were the blots quantified?

- We corrected this mistake (it is indeed 2.5fold). Lipids were added in PC-based liposomes containing the indicated phospholipid (now better explained in the figure legend). Lipids were obtained from Avanti Polar Lipids (egg PA and PC, brain PIP2) and Echelon (PIP3).

Preparation of liposomes is explained in the Methods section: "Liposomes (10mM total lipid concentration, either PC alone or PC:PA/PtdIns(3,4,5)P3/PtdIns(4,5)P2 in a 9:1 molar ratio) were prepared in LB buffer (30mM Tris, 4 mM EGTA, pH 8.0) by extrusion through a 0.1 μ m polycarbonate membrane using a Mini-Extruder (Avanti Polar Lipids, Inc.). The membrane floatation was performed as follows: liposomes (100 μ l of 10 mM total lipid concentration) were incubated on ice for 30 min with 1 μ g recombinant protein. LB buffer (30mM Tris, 4 mM EGTA, 2 M sucrose [pH 8.0]) was added to the incubation reaction to bring the final sucrose concentration to 1.6 M, and this mixture was overlaid with cushions containing 1.4M, 0.4M, and 0.25M sucrose in the same buffer in a TLA-55 tube. After centrifugation at 186,000 \times g (4 $^{\circ}$ C) for 45min in a TLA-55 rotor (Beckman), the 0.25/0.4 M interphase (top fraction, T) and the loading fraction (bottom fraction, B) were collected and analyzed by SDS-PAGE and Western Blot."

All experiments were performed as three independent experiments and band intensity (film exposure) was quantified using imageJ. We added a better explanation of the statistics and the missing information in the figure legend. We did not include PI5P as this lipid is predominately enriched in nuclear- and endoplasmic reticulum membranes (Jones et al. 2006) where LKB1 is not found at substantial amounts in vivo.

7) Fig. 4b - what do the asterisks signify? How were the statistical calculations performed? Multiple t-tests? Anova? Why are only some combinations examined? Were the experiments scored in a blinded fashion?

- The statistics were explained in the Methods section, which was unfortunately lost during the transfer process.

Statistics

All experiments were performed in triplicates. Error bars represent standard deviation and statistical significance was determined using ANOVA. $p < 0.0001$ **, $p < 0.001$ ***, $p < 0.01$ **, $p < 0.05$ *, $p > 0.05$ not significant (n.s.).**

We completed the asterisks for significance levels. Histology sections were scored in a blinded fashion.

8) Page 5, "Expression of PLD2 together with hLKB1 results in a further increase of AMPK activation (Fig. 4e), suggesting that PLD2-induced production of PA enhances LKB1 activity in vivo." - This is an interesting preliminary finding, but is nothing more than suggestive without further controls and extension of the work. What does PLD2 do by itself in this assay? What about transfecting a catalytically-inactive PLD2 allele? Does LKB1 require PLD activity to activate AMPK? There are very good PLD inhibitors developed by the Frohman and Brown groups that are commercially and inexpensively available from Sigma and other companies. Perhaps the PA is instead being generated by a DAG kinase? If LKB1 deficiency is embryonic lethal, why are mice lacking both PLD1 and PLD2 viable and relatively normal? - ***We thank the reviewer for these good suggestions. Indeed, transfection of PLD2 alone does not affect AMPK activation and expression of catalytically inactive PLD2 together with hLKB1 does not promote AMPK phosphorylation as its wild type counterpart does (Fig. 4F). Finally, inhibition of PLD2 by FIPI decreases LKB1-mediated AMPK activation (Fig. S5b). However, in vivo, other enzymes (e.g. DAG-kinase) may compensate for loss of PLD2 to produce PA. Thus, deletion of the lipid-binding motif in LKB1 does not result in the same phenotypes as knock-out of PLD in flies (viable), mice or worm (discussed now in the revised manuscript).***

9) Fig. 4g-k - interesting, but how was the quantitation done? Was it scored in a blinded fashion? There appears to be a lot of pS6 in the normal tissue as well. Finally, this sort of experiment is only correlative...

- ***A description of the scoring is explained in the methods section. In brief, sections were scored for no expression (0), weak (+), moderate (++) and strong expression (+++). Evaluation was performed in a blinded way. pS6K is indeed expressed at high levels in keratinocytes of the healthy skin but not in melanocytes (arrows in Fig. 4K). We agree (and mentioned in the manuscript) that these findings are only correlative – however combined with the in vitro, in cell culture and Drosophila in vivo data we believe to have a strong evidence for the described pathway.***

10) In the absence of better validation of the role of PLD (e.g. loss-of-function experiments), it is not possible to conclude (page 6) that the authors "have established here the first link between PLD-mediated production of PA and activation of the tumour suppressor LKB1." The PA, if important, could just as easily come from DAG kinase isoforms, or even PLD1. Showing that something happens with overexpression does not prove that that linkage occurs with endogenous proteins. For an example, see Y. Kanaho's work, PMID:23109426.

- ***We agree and changed the discussion in the revised manuscript. However, our main focus is the relation between PA and LKB1, which is indeed new and seems to be important for development and maybe the tumor suppressor function of LKB1. Furthermore, PLD2 is overexpressed in malignant melanoma samples, so in this particular pathological context we address the question of excess cellular PA and its consequences in the presence and absence of LKB1. Vice versa, decreased levels of PA (with a disturbed LKB1 activation) is indeed complicated, as other pathways might complement for a downregulation/loss of PLD (1, 2 and even 1+2) activity (as discussed in the revised manuscript, flies, worms and mice lacking PLDs are viable and exhibit rather minor phenotypes).***

11) Is there not supposed to be a materials and methods section in the paper? I could not find one. It is important to provide sources of key plasmids, antibodies, etc., as well as the methods that would be needed for others to attempt to reproduce the work.

- ***We apologize for that. The separate methods section file must have been lost during the transfer.***

Reviewer #2 (Remarks to the Author): Expert in drosophila neurodevelopment and cancer

In this manuscript, the authors attempt to demonstrate that the phospholipid binding by LKB1 is an upstream regulatory mechanism essential for LKB1 activity. The authors first showed that apart from its nuclear localizing inactive form, LKB1 could also localized to the cell membrane in mammalian and Drosophila epithelial cells as well as Drosophila embryonic neuroblasts. They discovered that the C-terminal polybasic stretch of amino acids is essential for its membrane localization particularly through interacting with phosphatidic acid. In addition, it was observed that the membrane binding deficient form of LKB1 exhibits impaired functionality in rescuing embryonic lethality and disturbed anterior-posterior polarity in Drosophila oocyte and axon formation in primary hippocampal neurons. Through kinase assay, the authors further showed that GFP-LKB1 Δ LB C564A displayed impaired kinase function, in particular with reduced AMPK phosphorylation. As a result, AMPK-mediated inhibition

on S6K phosphorylation and thus mTOR activation was reduced. LKB1-mediated cell viability was reduced when GFP-LKB1 Δ LB C564A was used. Interestingly, the authors showed that co-expression of PLD2 with LKB1 results in significant enhancement of LKB1-mediated phosphorylation of AMPK, which in turn leads to suppression of mTOR pathway. Lastly, the authors established a clinical relevance for their findings with human Melanoma, by demonstrating that a huge proportion of human Melanoma tumor exhibit reduced LKB1 level with increased PLD2 expression and Akt and mTOR activity. In a nutshell, this is a very novel and nice study and the authors have convincingly provided the first evidence on the membrane phospholipids binding domain of LKB1 and its importance for regulating LKB1 functionality. Furthermore, they have also provided evidence on the potential interaction between LKB1 and PLD, which is up-regulated in several cancer types. This manuscript will be of broad interest to readers of Nature Communications. I recommend publication of this manuscript provided that the authors are able to address the following concerns of the reviewer.

Major comments:

Mukhopadhyay et al 2015 demonstrated a reciprocal regulation of AMPK and PLD that increase in PLD activity reduced AMPK activity, while increase of AMPK activity reduced PLD activity. This paper needs to be cited. Mukhopadhyay et al 2015 showed that AICAR treatment increased p-AMPK, which is suppressed by addition of PA. However, authors of this manuscript showed that LKB1-induced phosphorylation of AMPK is enhanced by PLD2 expression. The authors need to explain the reason for these conflicting conclusions and probably provide additional evidence to support that PLD2 functions upstream to enhance the activity of LKB1.

- We added a new experiment (Fig. S5c) demonstrating that inhibition of PLD (by FIPI) decreases the activation of AMPK upon LKB1 transfection in our system. One difference between the experimental setup of Mukhopadhyay et al 2015 and our system is that we used an LKB1-deficient cell line (HeLa) whereas Mukhopadhyay et al used MDA-MB-231 cells, a highly dedifferentiated (almost mesenchymal) cell line derived from a mammary carcinoma metastasis, which expresses robust levels of endogenous LKB1 (Mukhopadhyay et al. 2015 and Linher-Melville and Singh 2014, PMID 24913037) but also high TGF. In contrast, LKB1 is frequently downregulated in breast cancer samples and cell lines derived from breast cancer metastases (Li et al. 2014, PMID 25178656), so MDA-MB-

231 cells might not reflect the in vivo situation of a typical breast cancer metastasis regarding the LKB1/AMPK signaling pathway. Furthermore, Mukhopadhyay et al. applied PA extracellularly. It has been demonstrated that this approach is capable to initiate some of the intracellular PA responses, however it might also affect other pathways than intracellularly produced PA (which is presumably only produced in distinct microdomains of the membrane). Finally, downregulation of PLD1+2 results in drastic effects in this particular cell line, whereas worms, flies and mice lacking both enzymes are viable and do not show dramatic phenotypes (LaLonde et al. 2005, PMID 15883198, Sato et al. 2013, PMID 23109426 and Raghun et al. 2009, PMID 19345277). Inhibition of PLD (new Fig. S5c) decreases the ability of LKB1 to activate AMPK only modestly, which is very likely only important under stressed conditions (e.g. in a tumor environment with high AMP levels).

Another point is that the effect observed by Mukhopadhyay et al. might be cell type specific. Beside HeLa cells (Fig. 4d-e and Fig. S5b), we included now an LKB1-deficient melanoma cell line (IGR37), which shows essentially the same results as HeLa (Fig. S6). We discuss these possible reasons for the differences to Mukhopadhyay et al. now in the new version of the manuscript.

Minor comments:

1. "However, overexpression of GFP-DmLKB1 Δ LB C564A in *lkb1*-mutant flies does not result in detectable rescue capacity (data not shown).

This data is already shown in Fig. 1o: the lethality/survival chart.

- Fig. 1o shows the rescue capacity of GFP-DmLKB1 Δ LB C564A expressed from its endogenous promoter. We added the data showing the overexpression (using the UAS/GAL4 system) of GFP-DmLKB1 Δ LB C564A in Fig. S3a).

2. "... or kinase-dead hLKB are inactive and fail to induce multiple axons (Fig. 4e)"

The kinase-dead hLKB1 data was not found in Fig 4.

- We apologize for this mistake and deleted this sentence. The function of LKB1 phosphorylating Sad kinase (which is likely to contribute to the multiple axon phenotype) has already been described previously (Barnes et al. 2007).

3. Fig 2H lacks a wt control.

*- The control (*lkb1::GFP-LKB1*) is shown in Fig. 1i,k,m. We refer now to this figure. For the sake of space, we did not include the wt expression also in Fig. 2.*

4. Colour scheme in Fig 1o can be improved.

- We changed the colors as requested.

5. In Fig 3d, GFP-LKB1 Δ LB C564A was almost undetectable, inconsistent with fig3c expression. This extremely low levels of protein will interfere with the conclusion for rescue.

- We added data for the rescue experiment (Figure S3a) with overexpressed GFP-DmLKB1 Δ LB C564A using the UAS/GAL4 system. Using this technique, we achieve protein levels similar to wild type GFP-LKB1 expressed from its endogenous promoter. However, GFP-DmLKB1 Δ LB C564A is still not capable to rescue the null allele. In Fig. 3c,

exposure settings are set very high to detect the residual protein expressed from the endogenous LKB1 promoter.

6. Fig 3g quantification does not correlate with kinase assay data.

- We apologize for this mistake and corrected the sentence (it must be 2.5fold instead of 4.5fold). As indicated in the Methods section (which was unfortunately lost during the transfer from NCB), the diagram represents the average results of three independent experiments. We also added one more experiment (Fig. S3b) using AMPK as substrate and readout (instead of STRAD phosphorylation) that confirmed the stimulation by PA.

Reviewer #3 (Remarks to the Author): Expert in AMPK signaling

A tumor suppressor protein kinase LKB1 plays diverse roles in various biological processes, including cell polarity, growth and also metabolism by regulating AMP-activated protein kinase (AMPK) and 12 other related kinases. The current manuscript claims that LKB1 is targeted to membrane through direct binding with phospholipids, which plays important role in fully activating AMPK and induction of multiple axons in primary neurons. However, there are several controls and descriptions of key experiments lacking and thus the authors claims are not compellingly supported by robust sets of data.

Major comments:

1. First of all, I could not find method section in the manuscript. Therefore, I am not able to judge if the experiments were done using standard/appropriate protocols. Moreover, there is no information regarding sample numbers (n), replicates, and descriptions of statistical analysis in the legend.

- We apologize for this – the Methods section was lost during the transfer from NCB. It is included now in the revised manuscript.

2. The authors have no comment (discussion) and measurement of AMPK-related kinases in this study. These kinases (e.g. SAD/BRISK, MARKs) have been shown in several studies that they play important roles in cell polarity and axon induction. It is necessary to show if at least some of those kinases is activated more strongly upon membrane localization of LKB1.

- We added new experiments showing that the activation of SadA (Fig. S5a) and MARK (Fig. 4E) depends on membrane-association of LKB1.

3. It is unclear the reason why membrane localization of LKB1 more robustly activate AMPK. Is it because AMPK and LKB1 co-localize at the membrane (as proposed by Bruce Kemp and Dario Alessi groups) or intrinsic activity of LKB1 is increased upon binding to phospholipids? Concerning the latter, measurement of endogenous LKB1 activity (IP kinase assay) should clarify this point.

- Our newly added experiment (Fig. S5b) demonstrates that impaired membrane binding does not result in a decreased association with the LKB1 cofactors STRAD α and Mo25 or one of its substrates (AMPK). Furthermore, Fig. 3f shows a significant decrease (to 25% compared to 100% for wild type LKB1) in kinase activity of LKB1 (immunoprecipitated from transfected cells). Vice versa, addition of PA-enriched liposomes enhances the kinase

activity of LKB1 (measured by STRAD α -phosphorylation and phosphorylation of AMPK in vitro, Fig. 3f and Fig. S3b).

Thus we think that binding to phospholipids (in particular to PA) is essential not only for membrane localization but also for the activation of the kinase function of LKB1 rather than membrane-binding serving as a platform for LKB1-substrate/cofactor assembly.

We discuss it now in the revised manuscript.

4. The authors used several point and truncation mutants of LKB1 in this study. It is necessary to show control experiments that such mutations/truncations do not alter intrinsic activity and/or binding to STRAD/MO25. There is no data showing when mutants were introduced to flies/cells, if they form function/stoichiometric complex with STRAD/MO25.

- We added a new figure (Fig. S3c) showing that membrane-binding-deficient LKB1 binds the same amount of STRAD α and Mo25 as its wild type counterpart.

5. Fig 3f, the authors describe that they measure LKB1 activity in vitro using STRAD as substrate. This is not a standard assay (no rationale) and the authors should show absolute activity (^{32}P incorporation into the substrate (STRAD)/min/mg).

- Fig. 3f represents an assay where functional LKB1 together with its cofactor STRAD α is immunoprecipitated from transfected cells. Subsequently, an in vitro kinase assay was performed and STRAD α phosphorylation determined as a readout (^{32}P incorporation was quantified by densitometric analysis of the relative intensity of the bands after exposure of gels to X-ray films using imageJ). As we immunoprecipitated the LKB1/STRAD complex, an exact determination of the amount of kinase complex (mg) was not possible. However, our loading control (Western Blotting) which was used for normalization of the radioactive signal indicated a very similar amount of proteins. Similar assays have been used previously by other groups (e.g. Lizcano et al. 2004). We agree that this is a semiquantitative approach, but the differences between wild type and membrane-binding deficient LKB1 should be obvious.

We furthermore added a kinase assay (Fig. S3b) using AMPK as substrate and readout (instead of STRAD phosphorylation) as suggested that confirmed the stimulation by PA.

Fig4e (right panel), the blots are not publication quality and higher pAMPK signal appears to be simply due to higher loading (based on total AMPK blot).

- We repeated the experiments to improve the quality of the blots and included them in Fig. 4e.

Reviewers' comments:

Reviewer #1 (Remarks to the Author):

The authors were responsive to my earlier comments. I have two more comments based on their changes to the manuscript.

1) In Fig. 3f, the authors purify LKB1/STRAD from S2R cells and show that the C564A mutant has a strongly decreased kinase activity relative to the wild-type kinase and conclude that membrane binding is important for the kinase activity. In this assay, there is no membrane present, right? Why would there be an effect of the C564A mutation? Does this suggest that there is a global misfolding effect of the mutation?

2) Why did the authors generate Y511F as a catalytically-inactive mutant of PLD2? The usual mutation is one to the HKD catalytic domain (the K of the second HKD). What is the evidence that this mutant is dead? The western blot in Fig. 4e in fact shows some effect of transfecting the Y511F mutant (quantitation would help here), suggesting that it is not actually catalytically inactive. The FIPI result in the supplemental figure is helpful - but the 4e result is not compelling.

Reviewer #2 (Remarks to the Author):

The authors have mostly addressed the reviewers' comments to my satisfaction.

But the careless mistakes is a bit concerning. In the author's rebuttal letter in response to my major comment, they wrongly referred their new data Fig S5b as fig S5c.

Regarding minor comment #3. Fig 2H lacks a wt control.

- The control (lkb1::GFP-LKB1) is shown in Fig. 1i,k,m. We refer now to this figure. For the sake of space, we did not include the wt expression also in Fig. 2."

If the author do not have space to show the wt control for Fig 2H, they should show them in the rebuttal letter. Presumably they have repeated the experiments and should have extra sets of images.

Reviewer #3 (Remarks to the Author):

Although the authors addressed some of my concerns, there are still serious issues remain unresolved. With the current data provided, I am still not convinced that PA activates LKB1.

-Measurement of LKB1 activity in vitro (Fig 3f and g)

I do not understand intention of the assay employed in Fig 3F. No one has done the assay in a way the authors had performed in the past. In addition, the assay procedure is poorly described in the Methods section and results are not clearly explained. According to the described method, the authors co-transfected LKB1 and STRAD with the same tag (OneStrep-GFP) and pulled down both components and assayed all together in vitro. Please describe that 1) such a big tag (OneStrep-GFP fusion) does not affect LKB1-STRAD-MO25 complex formation and activity, 2) how much lysates were used to purify tagged recombinant LKB1-STRAD, 3) what is the effect of endogenous LKB1-STRAD-MO25 binding to the preps in the in vitro assay?

Questions:

1) In Fig 3F, there is only one GFP band (just below 130 kDa marker) and the authors claim equal loading(?). Does the band represent STRAD or LKB1 or combination of both? If that is STRAD,

where is LKB1 loading control?

2) If you co-transfect LKB1 and STRAD together, it is possible that LKB1 had already phosphorylated STRAD prior to the pull-down and in vitro assay. It might be the case that STRAD was not further phosphorylated efficiently by the mutants because it had been phosphorylated in cells? I truly do not understand why the authors do not simply purify LKB1 (e.g. WT, mutants) complex and use well established LKB1 substrate to assay its activity in vitro (described in Lizcano JM et al EMBO J, 2004).

Then another thing that I do not understand is why the authors used commercial LKB1-STRAD-MO25 complex (from insect?) and assayed in the presence or absence of lipids rather using the same prep generated experiment shown in Fig 3g. It is not appropriate that the authors use different preps (different tag, different expression system/species, different complex (with/without MO25), from one experiment to the other without clear explanation. Please repeat the experiment using the same prep used in Fig 3f.

Supplementary Fig 3: What is "GFP-AMPK"? Is it human AMPK α 1 or α 2 or trimeric complex (e.g. α 1/ β 1/ γ 1)? AMPK expressed in cell culture must be already heavily phosphorylated and normally bacterial AMPK prep is used for in vitro assay (for substrate).

Point-to-point response to the Reviewers' comments:

Reviewer#1 (Remarks to the Author):

The authors were responsive to my earlier comments. I have two more comments based on their changes to the manuscript.

1) In Fig. 3f, the authors purify LKB1/STRAD from S2R cells and show that the C564A mutant has a strongly decreased kinase activity relative to the wild-type kinase and conclude that membrane binding is important for the kinase activity. In this assay, there is no membrane present, right? Why would there be an effect of the C564A mutation? Does this suggest that there is a global misfolding effect of the mutation?

- This is indeed an important point, we discuss it now in the revised version of the manuscript. We believe (as already discussed in the first revised version of manuscript) that membrane binding of LKB1 induces a conformational change of the protein, thereby activating the kinase domain. Whether the Δ LB+C564A mutation results in a global misfolding needs to be addressed in further studies, using distinct techniques such as crystallization and/or NMR and 3D-modeling, which is beyond the scope of this manuscript. Transfected wild type LKB1 is (even after purification, which is only a short period of ca. 1h due to the specificity and sensitivity of the OneStrep-tag) more active compared to membrane-binding-deficient LKB1. Furthermore, we have more evidences about the *in vivo* relevance of the membrane binding of LKB1:

1. Mammalian cell culture: In LKB1-deficient cells (HeLa or Melanoma cells, Fig. 4D, Fig. S5 and Fig. S6a) expression of wild type hLKB1 but not membrane-binding-deficient hLKB1 (hLKB1 $_{\Delta$ LB C430A}) induces an activation of AMPK, MARK and SAD.

2. *Drosophila*: Membrane-binding-deficient LKB1 is not capable to rescue an *lkb1* null allele (Fig. 1o and Fig. S3a).

2) Why did the authors generate Y511F as a catalytically-inactive mutant of PLD2? The usual mutation is one to the HKD catalytic domain (the K of the second HKD). What is the evidence that this mutant is dead? The western blot in Fig. 4e in fact shows some effect of transfecting the Y511F mutant (quantitation would help here), suggesting that it is not actually catalytically inactive. The FIPI result in the supplemental figure is helpful - but the 4e result is not compelling

- We apologize that we have not cited the used plasmid correctly (which is actually a matter of restricted amount of literature in Nat. Commun.). PLD2 Y511F was described by the group of Julian Gomez-Cambronero (Henkels et al. 2009, PMID19715678). It shows a reduction of PLD-activity by 80%, thus it is almost catalytically inactive. We added the reference in the revised version of the manuscript (page 10, line 3).

- We have added a quantification of pAMPK normalized against total AMPK in Fig. 4e, demonstrating that expression of PLD_{C1} alone or together with hLKB1 does not substantially affect AMPK phosphorylation.

Reviewer #2 (Remarks to the Author):

The authors have mostly addressed the reviewers' comments to my satisfaction.

But the careless mistakes is a bit concerning. In the author's rebuttal letter in response to my major comment, they wrongly referred their new data Fig S5b as fig S5c.

- We apologize for this mistake and correct it in the revised version of the manuscript.

Regarding minor comment #3. Fig 2H lacks a wt control.

- The control (Ikb1::GFP-LKB1) is shown in Fig. 1i,k,m. We refer now to this figure. For the sake of space, we did not include the wt expression also in Fig. 2." If the author do not have space to show the wt control for Fig 2H, they should show them in the rebuttal letter. Presumably they have repeated the experiments and should have extra sets of images.

- We agree and included the requested wild type control in Fig. 2h.

Reviewer #3 (Remarks to the Author):

Although the authors addressed some of my concerns, there are still serious issues remain unresolved. With the current data provided, I am still not convinced that PA activates LKB1.

-Measurement of LKB1 activity in vitro (Fig 3f and g)

I do not understand intention of the assay employed in Fig 3F. No one has done the assay in a way the authors had performed in the past. In addition, the assay procedure is poorly described in the Methods section and results are not clearly explained. According to the described method, the authors co-transfected LKB1 and STRAD with the same tag (OneStrep-GFP) and pulled down both components and assayed all together in vitro. Please describe that

1) such a big tag (OneStrep-GFP fusion) does not affect LKB1-STRAD-MO25 complex formation and activity,

- As frequently used in other studies (e.g. by the Alessi group), we pull down LKB1 and STRAD α . One-Strep is a very small tag (30aa) fused to the N-terminal of GFP, which we use to efficiently pull down the proteins. *Drosophila* GFP-LKB1 (Fig. 1o) is fully functional and associates with STRAD α /Stlk and Mo25 (Fig. S3). Other groups used GST-LKB1 (e.g. Lizcano *et al.* instead of OneStrep-GFP, which has more or less the same size. Therefore, we believe that OneS-GFP-LKB1 is fully active and can be used in these experiments.

2) how much lysates were used to purify tagged recombinant LKB1-STRAD,

- Using recombinant hLKB1-STRAD α -Mo25 complex, we used 1 μ g (as specified in the methods section). For experiments with *Drosophila* LKB1, the (wild type and mutant) proteins were pulled down from transfected S2R cells (2mg of total lysate).

We added this specification in the methods section:

In vitro kinase assay

OneStrep-GFP-LKB1 plus OneStrep-GFP-Stlk were precipitated from transfected S2R cells (2mg total protein lysates) using Streptactin beads (IBA, Goettingen, Germany).....

3) what is the effect of endogenous LKB1-STRAD-MO25 binding to the preps in the in vitro assay?

- We do not believe that this concern affects the outcome of the experiment as S2R cells express very low levels of endogenous LKB1 compared to the transfected GFP-LKB1 variants. This is supported by the finding that a kinase-dead variant of LKB1 exhibits a very low autophosphorylation and substrate phosphorylation, which is very similar to the control transfection (Fig. 3f).

Questions:

1) In Fig 3F, there is only one GFP band (just below 130 kDa marker) and the authors claim equal loading(?). Does the band represent STRAD or LKB1 or combination of both? If that is

STRAD, where is LKB1 loading control?

- **We exchanged the figure as we included now AMPK as a substrate. In the revised version, we show the full GFP blot, which shows GFP-LKB1 and GFP-STRAD α (Fig. 3f and g).**

2) If you co-transfect LKB1 and STRAD together, it is possible that LKB1 had already phosphorylated STRAD prior to the pull-down and *in vitro* assay. It might be the case that STRAD was not further phosphorylated efficiently by the mutants because it had been phosphorylated in cells?

- **This is of course to some extent possible, however phosphorylation and dephosphorylation are likely to be in an equilibrium, thus leaving enough unphosphorylated protein to be efficiently phosphorylated by LKB1 with radioactively labeled phosphate.**

In the revised version, we added another experiment with AMPK as substrate, which was recombinantly expressed in bacteria and thus cannot be phosphorylated prior to the assay. The differences in AMPK-phosphorylation are comparable to the LKB1 autophosphorylation levels (Fig. 3f).

Furthermore, our mammalian cell culture transfections using a phospho-specific antibody against AMPK (LKB1 phosphorylation site in the T-loop) clearly demonstrate that membrane-binding deficient LKB1 is almost not able of AMPK phosphorylation/activation (Fig. 4d and Fig. S6a).

I truly do not understand why the authors do not simply purify LKB1 (e.g. WT, mutants) complex and use well established LKB1 substrate to assay its activity *in vitro* (described in Lizcano JM et al EMBO J, 2004).

- **We intensively tried to purify recombinant LKB1 variants from *E.coli* and succeeded to get enough protein for immunization and biochemical assays, however this protein exhibits a very low catalytic activity. Thus we moved to purification from transfected cells, which is in our eyes a very defined approach to address the proteins functionality (and e.g. used by Dario Alessi's group in Boudeau et al. 2003, PMID14517248, Hans Clever's group in Baas et al. 2003, PMID12805220 and others).**

We agree that using a physiological substrate is a more appropriate way to evaluate the kinase activity. Therefore, we performed new *in vitro* kinase experiments (new Fig. 3f and g) with *Drosophila* LKB1 purified from transfected S2R cells and a recombinantly expressed fragment of AMPK (GST-AMPK108-280). We now show LKB1 autophosphorylation and AMPK phosphorylation in the new figure (Fig. 3f and g). The results are essentially the same, demonstrating a strong decrease in the *in vitro* kinase activity if the membrane-binding capacity of LKB1 is abolished (Fig. 3f) and vice versa an increase in LKB1 (wild type but not Δ LB C564A) kinase activity when PA-enriched liposomes are added to the reaction (Fig. 3g).

Furthermore, we show *in vivo* that in LKB1-deficient cells (HeLa or Melanoma cells, Fig. 4D and Fig. S6a), expression of wild type LKB1 but not membrane-binding deficient LKB1 (hLKB1 Δ LB C430A) induces an activation of AMPK and MARK (and SAD, Fig. S5).

Finally, membrane-binding deficient is neither capable of rescuing an *lkb1* null allele (Fig. 1o and Fig. S3a) nor of inducing axonal polarity (Fig. 5).

Therefore, we believe that all these experiments strongly substantiate our hypothesis that membrane binding of LKB1 (and in particular binding to phosphatidic acid) is essential for the proteins functionality.

Then another thing that I do not understand is why the authors used commercial LKB1-STRAD-MO25 complex (from insect?) and assayed in the presence or absence of lipids rather using the same prep generated experiment shown in Fig 3g. It is not appropriate that the authors use different preps (different tag, different expression system/species, different complex (with/without MO25), from one experiment to the other without clear explanation. Please repeat the experiment using the same prep used in Fig 3f.

- We apologize for this confusion. We have added new experiments (new Fig. 3f and g): In the revised version, figure 3 describes only *Drosophila* proteins, which have been purified from transfected S2R cells (as discussed above).

We found the described mechanism to be conserved in human LKB1 in separated figures (Fig. S3: *in vitro* kinase assay with addition of liposomes, Fig. 4: HeLa cell culture model for AMPK and MARK-phosphorylation, Fig. S5: SAD-phosphorylation in HeLa cell culture model, Fig. S6: AMPK and MARK-phosphorylation in melanoma cell lines).

Supplementary Fig 3: What is “GFP-AMPK”? Is it human AMPK α 1 or α 2 or trimeric complex (e.g. α 1/ β 1/ γ 1)? AMPK expressed in cell culture must be already heavily phosphorylated and normally bacterial AMPK prep is used for *in vitro* assay (for substrate).

- We failed to purify bacterially expressed full length AMPK. We were not able to use a smaller GST-AMPK fragment, as the signal would have interfered with the autophosphorylation band of the hLKB1-hMo25-hSTRAD α -complex. Instead, we expressed human GFP-AMPK α 1, which shows no intrinsic kinase activity (Fig. S3b, first lane). Although it might have been phosphorylated in HeLa cells (but not by hLKB1 as HeLa do not express detectable level of hLKB1), in this radioactive kinase assay we visualize only the new phosphorylation by recombinant hLKB1.

We specify it now better in the figure legend and material and methods section:

“Supplementary Figure 3: Lipid-binding of LKB1 is essential for its function.

.....(c) Kinase activity of recombinant hLKB1/hSTRAD α /hMo25 is strongly increased by addition of PC + PA, but only slightly by PC, PC + PtdIns(4,5)P2 or PC + PtdIns(3,4,5)P3. Lipids were prepared as liposomes as described in methods sections and applied in a 9:1 (PC:X) molar ratio. (d) *In vitro* kinase assays of GFP-AMPK α 1 purified from transfected HeLa cells using recombinant hLKB1/STRAD α /Mo25 and indicated lipids. GFP-AMPK phosphorylation in the absence of LKB1 was not detectable. Phosphorylation of GFP-AMPK by LKB1 in the absence of lipids was set as 100%.”

Methods

In vitro kinase assay

.....

For assays with recombinant kinase complex (Fig S3b-c), 1 μ g of a complex of recombinant hLKB1, STRAD α and Mo25 (SIGMA) was used as described above replacing immunoprecipitated LKB1 protein. In this experiment, GFP-AMPK α 1 from transfected HeLa cells (2mg total protein lysate) was used as substrate as a recombinant fragment would have interfered with the autophosphorylation bands. After isolation of precipitated OneStrep-GFP-AMPK, recombinant hLKB1/STRAD α /Mo25 was added and incubated as described above. Subsequently, OneStrep-GFP-AMPK was purified from the reaction mixture using Streptactin beads.

For kinase assays with addition of lipids, Liposomes were prepared as described above and added at 10mM final concentration.

REVIEWERS' COMMENTS:

Reviewer #1 (Remarks to the Author):

Unfortunately, neither of my concerns were addressed in the response to the prior version of the manuscript.

The point I was raising in reference to the C564A mutant having much less activity than the wild-type protein in an in vitro assay in which there is no membrane (Fig. 3f) is that this indicates that there is more altered in the behavior of the enzyme than its inability to bind membrane, and hence one can't conclude that it is the loss of membrane binding that is responsible for its lack of inactivation in cells. In essence, this finding indicates that conclusions can't be drawn using this tool.

Similarly, my concern in pointing out the mis-use of Y511F PLD2 is that it is not catalytically inactive, as the authors now acknowledge. In Fig 4e, I agree that the pAMPK does not look substantially increased by Y511F, but a substantial increase in pMARK is pretty obvious and not acknowledged... The figure, figure legend, and text still describe the isoform as PLD2ci, which is not acceptable since it is not an inactive enzyme. The appropriate response would have been to re-do any experiments that had used Y511F with an actually catalytically-dead enzyme. Although one could re-write the text and interpretation to discuss the effects of a control with partial activity that didn't rescue pAMPK but did rescue pMARK, that wouldn't be a good control nor would be results be compelling.

Reviewer #2 (Remarks to the Author):

I now support the publication of this revised manuscript.

Reviewer #3 (Remarks to the Author):

Although the authors were not able to perform the assays in a more optional condition (for multiple assays) I had asked, they used alternative assay conditions and addressed my concerns at minimal, but acceptable level.

Reviewer #1 (Remarks to the Author):

Unfortunately, neither of my concerns were addressed in the response to the prior version of the manuscript.

The point I was raising in reference to the C564A mutant having much less activity than the wild-type protein in an *in vitro* assay in which there is no membrane (Fig. 3f) is that this indicates that there is more altered in the behavior of the enzyme than its inability to bind membrane, and hence one can't conclude that it is the loss of membrane binding that is responsible for its lack of inactivation in cells. In essence, this finding indicates that conclusions can't be drawn using this tool.

- **We already discussed in the revised version of the manuscript, that we are convinced that membrane binding induces a conformational change of LKB1, thus activating its kinase domain. Indeed, micelles co-immunoprecipitated with the LKB1/STRAD α -complex harvested from S2R cells might be the reason why the kinase activity in wild type LKB1 is enhanced even in *in vitro* kinase assay. We pointed that out in the revised version of the manuscript (Discussion section):**
- "In our *in vitro* kinase assays using immunoprecipitated proteins from transfected cells wild type LKB1 might have co-immunoprecipitated with PA-enriched micelles, which enhanced its kinase activity in this experiment."

Similarly, my concern in pointing out the mis-use of Y511F PLD2 is that it is not catalytically inactive, as the authors now acknowledge. In Fig 4e, I agree that the pAMPK does not look substantially increased by Y511F, but a substantial increase in pMARK is pretty obvious and not acknowledged... The figure, figure legend, and text still describe the isoform as PLD2ci, which is not acceptable since it is not an inactive enzyme. The appropriate response would have been to re-do any experiments that had used Y511F with an actually catalytically-dead enzyme. Although one could re-write the text and interpretation to discuss the effects of a control with partial activity that didn't rescue pAMPK but did rescue pMARK, that wouldn't be a good control nor would be results be compelling.

- **PLD Y511F has been described to be almost a catalytically inactive variant: Henkels et al. 2009, PMID: 19715678, Figure 2A, PLD2 Y511F increased the PLD activity ca. 30% over mock transfection, whereas wild type PLD exhibit a ca. 450% increase. Nonetheless, we changed the annotation to "catalytically reduced PLD2".**

Reviewer #2 (Remarks to the Author):

I now support the publication of this revised manuscript.

Reviewer #3 (Remarks to the Author):

Although the authors were not able to perform the assays in a more optional condition (for multiple assays) I had asked, they used alternative assay conditions and addressed my concerns at minimal, but acceptable level.